# Function Space Bayesian Pseudocoreset for Bayesian Neural Networks

**Balhae Kim**[1], **Hyungi Lee**[1], **Juho Lee**[1,2]
KAIST AI[1], AITRICS[2]
{balhaekim, lhk2708, juholee}@kaist.ac.kr

## Abstract

A Bayesian pseudocoreset is a compact synthetic dataset summarizing essential information of a large-scale dataset and thus can be used as a proxy dataset for scalable Bayesian inference. Typically, a Bayesian pseudocoreset is constructed by minimizing a divergence measure between the posterior conditioning on the pseudocoreset and the posterior conditioning on the full dataset. However, evaluating the divergence can be challenging, particularly for the models like deep neural networks having high-dimensional parameters. In this paper, we propose a novel Bayesian pseudocoreset construction method that operates on a function space. Unlike previous methods, which construct and match the coreset and full data posteriors in the space of model parameters (weights), our method constructs variational approximations to the coreset posterior on a function space and matches it to the full data posterior in the function space. By working directly on the function space, our method could bypass several challenges that may arise when working on a weight space, including limited scalability and multi-modality issue. Through various experiments, we demonstrate that the Bayesian pseudocoresets constructed from our method enjoys enhanced uncertainty quantification and better robustness across various model architectures.

## 1 Introduction

Deep learning has achieved tremendous success, but its requirement for large amounts of data makes it often inefficient or infeasible in terms of resources and computation. To enable continuous learning like humans, it is necessary to learn from a large number of data points in a continuous manner, which requires the ability to discern and retain important information. This motivates the learning of a coreset, a small dataset that is informative enough to represent a large dataset.

On the other hand, the ability to incorporate uncertainties into predictions is essential for real-world applications, as it contributes to the safety and reliability of a model. One approach to achieving this is by adopting a Bayesian framework, where a prior distribution is established to represent our initial belief about the models. This belief is then updated through inference of posterior distributions based on the acquired knowledge. Although this approach shows promise, scalability becomes a concern when working with large-scale datasets during Bayesian inference. To address this issue, a potential solution is to employ a Bayesian coreset. A Bayesian coreset is a small subset of the original dataset where the posterior conditioning on it closely approximates the original posterior conditioning on the full dataset. Once the Bayesian coreset is trained, it can be utilized as a lightweight proxy dataset for subsequent Bayesian inference or as a replay buffer for continual learning or transfer learning.

A Bayesian coreset is constructed by selecting a subset from a large dataset. However, recent research suggests that this approach may not be effective, especially in high-dimensional settings [19]. Instead, an alternative method of synthesizing a coreset, wherein the coreset is learned as trainable parameters, has been found to significantly enhance the quality of the approximation. This synthesized coreset is

37th Conference on Neural Information Processing Systems (NeurIPS 2023).

referred to as a Bayesian pseudocoreset. The process of learning a Bayesian pseudocoreset involves minimizing a divergence measure between the posterior of the full dataset and the posterior of the pseudocoreset. However, learning Bayesian pseudocoresets is generally challenging due to the intractability of constructing both the full dataset posterior and the pseudocoreset posterior, as well as the computation of the divergence between them, which necessitates approximation. Consequently, existing works on Bayesian pseudocoresets have primarily focused on small-scale problems [19, 8, 20, 21]. Recently, Kim et al. [15] introduced a scalable method for constructing Bayesian pseudocoresets using variational Gaussian approximation for the posteriors and minimizing forward KL divergence. Although their method shows promise, it still demands substantial computational resources for high-dimensional models like deep neural networks.

In this paper, we present a novel approach to enhance the scalability of Bayesian pseudocoreset construction, particularly for Bayesian neural networks (BNNs) with a large number of parameters. Our proposed method operates in *function space*. When working with BNNs, it is common to define a prior distribution on the weight space and infer the corresponding weight posterior distribution, which also applies to Bayesian pseudocoreset construction. However, previous studies [31, 28] have highlighted the challenge of interpreting weights in high-dimensional neural networks, making it difficult to elicit meaningful prior distributions. Additionally, in high-dimensional networks, the loss surfaces often exhibit a complex multimodal structure, which means that proximity in the weight space does not necessarily imply proximity in the desired prediction variable [24, 30]. This same argument can be applied to Bayesian pseudocoreset construction, as matching the full data and pseudocoreset posteriors in the weight space may not result in an optimal pseudocoreset in terms of representation power and computational scalability.

To be more specific, our method constructs a variational approximation to the pseudocoreset posteriors in function space by linearization and variational approximation to the true posterior. Then we learn Bayesian pseudocoreset by minimizing a divergence measure between the full data posterior and the pseudocoreset posterior in the function space. Compared to the previous weight space approaches, our method readily scales to the large models for which the weight space approaches were not able to compute. Another benefit of function space matching is that it does not constrain the architectures of the neural networks to be matched, provided that their inherited function space posteriors are likely to be similar. So for instance, we can train with multiple neural network architectures simultaneously with varying numbers of neurons or types of normalization layers, and we empirically observe that this improves the architectural robustness of the learned pseudocoresets. Moreover, it has another advantage that the posteriors learned from the Bayesian pseudocoreset in function space have better out-of-distribution (OOD) robustness, similar to the previous reports showing the benefit of function space approaches in OOD robustness [28].

In summary, this paper presents a novel approach to creating a scalable and effective Bayesian pseudocoreset using function space variational inference. The resulting Bayesian pseudocoreset is capable of being generated in high-dimensional image and deep neural network settings and has better uncertainty quantification abilities compared to weight space variational inference. Additionally, it has better architectural robustness. We demonstrate the efficiency of the function space Bayesian pseudocoreset through the various experiments.

## 2 Background

### 2.1 Bayesian pseudocoresets

In this paper, we focus on probabilistic models for supervised learning problem. Let $\theta \in \Theta$ be a parameter, and let $p(y \,|\, x, \theta)$ be a probabilistic model indexed by the parameter $\theta$. Given a set of observations $\mathbf{x} := (x_i)_{i=1}^n$ and the set of labels $\mathbf{y} := (y_i)_{i=1}^n$ with each $x_i \in \mathcal{X}$ and $y_i \in \mathcal{Y}$, we are interested in updating our prior belief $\pi_0(\theta)$ about the parameter to the posterior,

$$\pi_{\mathbf{x}}(\theta) := \frac{\pi_0(\theta)}{Z(\mathbf{y} \,|\, \mathbf{x})} \prod_{i=1}^n p(y_i \,|\, x_i, \theta), \quad Z(\mathbf{y} \,|\, \mathbf{x}) := \int_\Theta \prod_{i=1}^n p(y_i \,|\, x_i, \theta) \pi_0(\mathrm{d}\theta). \tag{1}$$

However, when the size of the dataset $n$ is large, the computation of the posterior distribution can be computationally expensive and infeasible. To overcome this issue, Bayesian pseudocoresets are constructed as a synthetic dataset $\mathbf{u} = (u_j)_{j=1}^m$ with $m \ll n$ with the set of labels $\tilde{\mathbf{y}} := (\tilde{y}_j)_{j=1}^m$

where the posterior conditioning on it approximates the original posterior $\pi_{\mathbf{x}}(\theta)$.

$$\pi_{\mathbf{u}}(\theta) = \frac{\pi_0(\theta)}{Z(\tilde{\mathbf{y}} \mid \mathbf{u})} \prod_{j=1}^{m} p(\tilde{y}_j \mid u_j, \theta), \quad Z(\tilde{\mathbf{y}} \mid \mathbf{u}) := \int_{\Theta} \prod_{j=1}^{m} p(\tilde{y}_j \mid u_j, \theta) \pi_0(\mathrm{d}\theta). \tag{2}$$

This approximation is made possible by solving an optimization problem that minimizes a divergence measure $D$ between the two posterior distributions [1]

$$\mathbf{u}^* = \arg\min_{\mathbf{u}} \; D(\pi_{\mathbf{x}}, \pi_{\mathbf{u}}). \tag{3}$$

In a recent paper [15], three variants of Bayesian pseudocoresets were proposed using different divergence measures, namely reverse Kullback-Leibler divergence, Wasserstein distance, and forward Kullback-Leibler divergence. However, performing both the approximation and the optimization in the parameter space can be computationally challenging, particularly for high-dimensional models such as deep neural networks.

## 2.2 Bayesian pseudocoresets in weight-space

Kim et al. [15] advocates using forward KL divergence as the divergence measure when constructing Bayesian pseudocoresets, with the aim of achieving a more even exploration of the posterior distribution of the full dataset when performing uncertainty quantification with the learned pseudocoreset. The forward KL objective is computed as,

$$D_{\mathrm{KL}}[\pi_{\mathbf{x}} \| \pi_{\mathbf{u}}] = \log Z(\tilde{\mathbf{y}} \mid \mathbf{u}) - \log Z(\mathbf{y} \mid \mathbf{x})$$
$$+ \mathbb{E}_{\pi_{\mathbf{x}}}\left[ \sum_{i=1}^{n} \log p(y_i \mid x_i, \theta) \right] - \mathbb{E}_{\pi_{\mathbf{x}}}\left[ \sum_{j=1}^{m} \log p(\tilde{y}_j \mid u_j, \theta) \right]. \tag{4}$$

The derivative of the divergence with respect to the pseudocoreset $\mathbf{u}$ is computed as

$$\nabla_{\mathbf{u}} D_{\mathrm{KL}}[\pi_{\mathbf{x}} \| \pi_{\mathbf{u}}] = \mathbb{E}_{\pi_{\mathbf{u}}}\left[ \nabla_{\mathbf{u}} \sum_{j=1}^{m} \log p(\tilde{y}_j \mid u_j, \theta) \right] - \nabla_{\mathbf{u}} \mathbb{E}_{\pi_{\mathbf{x}}}\left[ \sum_{j=1}^{m} \log p(\tilde{y}_j | u_j, \theta)) \right] \tag{5}$$

For the gradient, we need the expected gradients of the log posteriors that require sampling from the posteriors $\pi_{\mathbf{x}}$ and $\pi_{\mathbf{u}}$. Most of the probabilistic models do not admit simple closed-form expressions for these posteriors, and it is not easy to simulate those posteriors for high-dimensional models. To address this, Kim et al. [15] proposes to use a Gaussian variational distributions $q_{\mathbf{u}}(\theta)$ and $q_{\mathbf{x}}(\theta)$ to approximate $\pi_{\mathbf{x}}$ and $\pi_{\mathbf{u}}$ whose means are set to the parameters obtained from the SGD trajectories,

$$q_{\mathbf{u}}(\theta) = \mathcal{N}(\theta; \mu_{\mathbf{u}}, \Sigma_{\mathbf{u}}), \quad q_{\mathbf{x}}(\theta) = \mathcal{N}(\theta; \mu_{\mathbf{x}}, \Sigma_{\mathbf{x}}), \tag{6}$$

where $\mu_{\mathbf{u}}$ and $\mu_{\mathbf{x}}$ are the maximum a posteriori (MAP) solutions computed for the dataset $\mathbf{u}$ and $\mathbf{x}$, respectively. $\Sigma_{\mathbf{u}}$ and $\Sigma_{\mathbf{x}}$ are covariances. The gradient, with the stop gradient applied to $\mu_{\mathbf{u}}$, is approximated as,

$$\frac{\nabla_{\mathbf{u}}}{S} \sum_{s=1}^{S} \left( \sum_{j=1}^{m} \log p\left( \tilde{y}_j \mid u_j, \mathbf{sg}(\mu_{\mathbf{u}}) + \Sigma_{\mathbf{u}}^{1/2} \varepsilon_{\mathbf{u}}^{(s)} \right) - \sum_{j=1}^{m} \log p\left( \tilde{y}_j \mid u_j, \mu_{\mathbf{x}} + \Sigma_{\mathbf{x}}^{1/2} \varepsilon_{\mathbf{x}}^{(s)} \right) \right). \tag{7}$$

Here, $\varepsilon_{\mathbf{u}}^{(s)}$ and $\varepsilon_{\mathbf{x}}^{(s)}$ are i.i.d. standard Gaussian noises and $S$ is the number of Monte-Carlo samples.

**Expert trajectories** Approximating the full data and coreset posteriors with variational distributions as specified above requires $\mu_{\mathbf{u}}$ and $\mu_{\mathbf{x}}$ as consequences of running optimization algorithms untill convergence. While this may be feasible for small datasets, for large-scale setting of our interest, obtaining $\mu_{\mathbf{u}}$ and $\mu_{\mathbf{x}}$ from scratch at each iteration for updating $\mathbf{u}$ can be time-consuming. To alleviate this, in the dataset distillation literature, Cazenavette et al. [7] proposed to use the *expert trajectories*, the set of pretrained optimization trajectories constructed in advance to the coreset learning. Kim et al. [15] brought this idea to Bayesian pseudocoresets, where a pool of pretrained trajectories are assume to be given before pseudocoreset learning. At each step of pseudocoreset update, a checkpoint $\theta_0$ from an expert trajectory is randomly drawn from the pool, and then $\mu_{\mathbf{u}}$ and $\mu_{\mathbf{x}}$ are quickly constructed by taking few optimization steps from $\theta_0$.

---

[1]In principle, we should learn the (pseudo)labels $\tilde{\mathbf{y}}$ as well, but for classification problem, we can simply fix it as a constant set containing equal proportion of all possible classes. We assume this throughout the paper.

# 3 Function space Bayesian pseudocoreset

## 3.1 Function space Bayesian neural networks

We follow the framework presented in Rudner et al. [29, 28] to define a Function-space Bayesian Neural Network (FBNN). Let $\pi_0(\theta)$ be a prior distribution on the parameter and $g_\theta : \mathcal{X} \to \mathbb{R}^d$ be a neural network index by $\theta$. Let $h : \Theta \to (\mathcal{X} \to \mathbb{R}^d)$ be a deterministic mapping from a parameter $\theta$ to a neural network $g_\theta$. Then a function-space prior is simply defined as a pushforward $\nu_0(f) = h_*\pi_0(f) := \pi_0(h^{-1}(f))$. The corresponding posterior is also defined as a pushforward $\nu_{\mathbf{x}}(f) = h_*\pi_{\mathbf{x}}(f)$ and so is the pseudocoreset posterior $\nu_{\mathbf{u}}(f) = h_*\pi_{\mathbf{u}}(f)$.

## 3.2 Learning function space Bayesian pseudocoresets

Given the function space priors and posteriors, a Function space Bayesian PseudoCoreset (FBPC) is obtained by minimizing a divergence measure between the function space posteriors,

$$\mathbf{u}^* = \arg\min_{\mathbf{u}} \ D(\nu_{\mathbf{x}}, \nu_{\mathbf{u}}). \tag{8}$$

We follow Kim et al. [15] suggesting to use the forward KL divergence, so our goal is to solve

$$\mathbf{u}^* = \arg\min_{\mathbf{u}} \ D_{\mathrm{KL}}[\nu_{\mathbf{x}}\|\nu_{\mathbf{u}}]. \tag{9}$$

The following proposition provides an expression for the gradient to minimize the divergence, whose proof is given Appendix A.

**Proposition 3.1.** *The gradient of the forward KL divergence with respect to the coreset $\mathbf{u}$ is*

$$\nabla_{\mathbf{u}} D_{\mathrm{KL}}[\nu_{\mathbf{x}}\|\nu_{\mathbf{u}}] = -\nabla_{\mathbf{u}}\mathbb{E}_{[\nu_{\mathbf{x}}]_{\mathbf{u}}}[\log p(\tilde{\mathbf{y}} \,|\, \mathbf{f}_{\mathbf{u}})] + \mathbb{E}_{[\nu_{\mathbf{u}}]_{\mathbf{u}}}[\nabla_{\mathbf{u}} \log p(\tilde{\mathbf{y}} \,|\, \mathbf{f}_{\mathbf{u}})], \tag{10}$$

*where $[\nu_{\mathbf{x}}]_{\mathbf{u}}$ and $[\nu_{\mathbf{u}}]_{\mathbf{u}}$ are finite-dimensional distributions of the stochastic processes $\nu_{\mathbf{x}}$ and $\nu_{\mathbf{u}}$, respectively, $\mathbf{f}_{\mathbf{u}} := (f(u_j))_{j=1}^m$, and $p(\tilde{\mathbf{y}} \,|\, \mathbf{f}_{\mathbf{u}}) = \prod_{j=1}^m p(\tilde{y}_j \,|\, f(u_j))$.*

To evaluate the gradient Eq. 10, we should identify the finite-dimensional functional posterior distributions $[\nu_{\mathbf{x}}]_{\mathbf{u}}$ and $[\nu_{\mathbf{u}}]_{\mathbf{u}}$. While this is generally intractable, as proposed in Rudner et al. [29, 28], we can instead consider a linearized approximation of the neural network $g_\theta$,

$$\tilde{g}_\theta(\cdot) = g_{\mu_{\mathbf{x}}}(\cdot) + \mathcal{J}_{\mu_{\mathbf{x}}}(\cdot)(\theta - \mu_{\mathbf{x}}), \tag{11}$$

where $\mu_{\mathbf{x}} = \mathbb{E}_{\pi_{\mathbf{x}}}[\theta]$ and $\mathcal{J}_{\mu_{\mathbf{x}}}(\cdot)$ is the Jacobian of $g_\theta$ evaluated at $\mu_{\mathbf{x}}$. Then we approximate the function space posterior $\nu_{\mathbf{x}}$ with $\tilde{\nu}_{\mathbf{x}} := \tilde{h}_*\pi_{\mathbf{x}}$ where $\tilde{h}(\theta) = \tilde{g}_\theta$, and as shown in Rudner et al. [29, 28], the finite dimensional distribution $[\tilde{\nu}_{\mathbf{x}}]_{\mathbf{u}}$ is a multivariate Gaussian distribution,

$$[\tilde{\nu}_{\mathbf{x}}]_{\mathbf{u}}(\mathbf{f}_{\mathbf{u}}) = \mathcal{N}\Big(\mathbf{f}_{\mathbf{u}} \,|\, g_{\mu_{\mathbf{x}}}(\mathbf{u}), \mathcal{J}_{\mu_{\mathbf{x}}}(\mathbf{u})\Sigma_{\mathbf{x}}\mathcal{J}_{\mu_{\mathbf{x}}}(\mathbf{u})^\top\Big), \tag{12}$$

with $\Sigma_{\mathbf{x}} = \mathrm{Cov}_{\pi_{\mathbf{x}}}(\theta)$. Similarly, we obtain

$$[\tilde{\nu}_{\mathbf{u}}]_{\mathbf{u}}(\mathbf{f}_{\mathbf{u}}) = \mathcal{N}\Big(\mathbf{f}_{\mathbf{u}} \,|\, g_{\mu_{\mathbf{u}}}(\mathbf{u}), \mathcal{J}_{\mu_{\mathbf{u}}}(\mathbf{u})\Sigma_{\mathbf{u}}\mathcal{J}_{\mu_{\mathbf{u}}}(\mathbf{u})^\top\Big), \tag{13}$$

with $\mu_{\mathbf{u}} := \mathbb{E}_{\pi_{\mathbf{u}}}[\theta]$ and $\Sigma_{\mathbf{u}} := \mathrm{Cov}_{\pi_{\mathbf{u}}}(\theta)$. Using these linearized finite-dimensional distribution, we can approximate

$$\nabla_{\mathbf{u}} D_{\mathrm{KL}}[\nu_{\mathbf{x}}\|\nu_{\mathbf{u}}] = -\nabla_{\mathbf{u}}\mathbb{E}_{[\tilde{\nu}_{\mathbf{x}}]_{\mathbf{u}}}[\log p(\tilde{\mathbf{y}} \,|\, \mathbf{f}_{\mathbf{u}})] + \mathbb{E}_{[\tilde{\nu}_{\mathbf{u}}]_{\mathbf{u}}}[\nabla_{\mathbf{u}} \log p(\tilde{\mathbf{y}} \,|\, \mathbf{f}_{\mathbf{u}})], \tag{14}$$

## 3.3 Tractable approximation to the gradient

Even with the linearization, evaluating Eq. 14 is still challenging because it requires obtaining $\mu_{\mathbf{x}}$ and $\Sigma_{\mathbf{x}}$ which are the statistics of the weight-space posterior $\pi_{\mathbf{x}}$. Rudner et al. [28] proposes to learn a variational approximation $q_{\mathbf{x}}(\theta)$ in the *weight-space*, and use the linearized pushforward of the variational distribution $\tilde{h}_*q_{\mathbf{x}}$ as a proxy to the function space posterior. Still, this approach requires computing the heavy Jacobian matrix $\mathcal{J}_{\mathbb{E}_{q_{\mathbf{x}}[\theta]}}(\mathbf{u})$, so may not be feasible for our scenario where we have to compute such variational approximations *at each* update of the pseudocoreset $\mathbf{u}$.

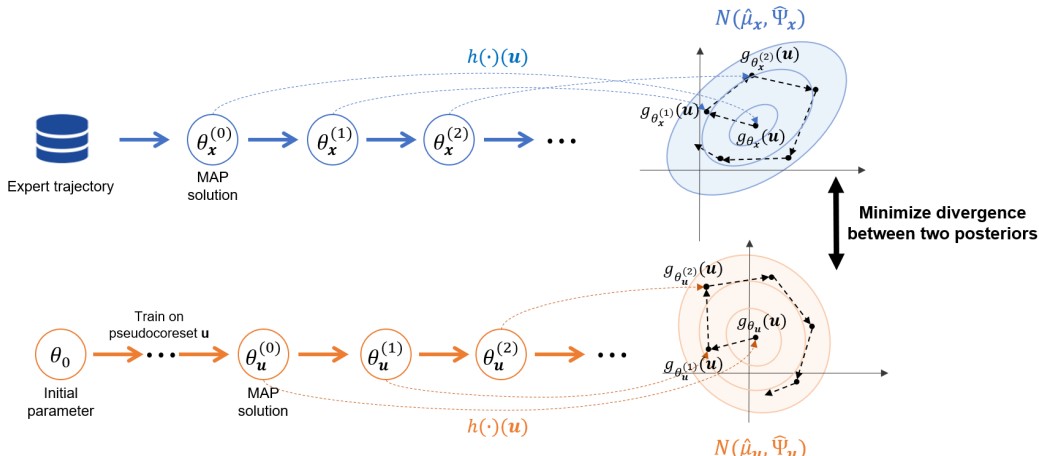

**Figure 1:** The conceptual overview of our proposed training procedure.

Instead, we choose to directly construct a variational approximations to the finite-dimensional distributions of the function space posteriors, that is,

$$
\begin{aligned}
[\tilde{\nu}_{\mathbf{x}}]_{\mathbf{u}}(\mathbf{f}_{\mathbf{u}}) &\approx q_{\mathbf{x}}(\mathbf{f}_{\mathbf{u}}) = \mathcal{N}(\mathbf{f}_{\mathbf{u}} \,|\, g_{\hat{\mu}_{\mathbf{x}}}(\mathbf{u}), \hat{\Psi}_{\mathbf{x}}), \\
[\tilde{\nu}_{\mathbf{u}}]_{\mathbf{u}}(\mathbf{f}_{\mathbf{u}}) &\approx q_{\mathbf{u}}(\mathbf{f}_{\mathbf{u}}) = \mathcal{N}(\mathbf{f}_{\mathbf{u}} \,|\, g_{\hat{\mu}_{\mathbf{u}}}(\mathbf{u}), \hat{\Psi}_{\mathbf{u}}),
\end{aligned}
\tag{15}
$$

where $(\hat{\mu}_{\mathbf{x}}, \hat{\Psi}_{\mathbf{x}})$ and $(\hat{\mu}_{\mathbf{u}}, \hat{\Psi}_{\mathbf{u}})$ are variational parameters for the full data and coreset posteriors. Inspired by Kim et al. [15], we construct the variational parameters using expert trajectories. Unlike [15], we simply let the MAP solution computed for $\mathbf{x}$, $\theta_{\mathbf{x}}$, by sampling a checkpoint from the later part of the expert trajectories, and obtain the MAP solution of $\mathbf{u}$, $\theta_{\mathbf{u}}$, by directly optimizing an initial random parameter. Then we obtain $\hat{\mu}_{\mathbf{x}}$ and $\hat{\mu}_{\mathbf{u}}$ using $\mathbf{u}$. For the covariance matricies $\hat{\Psi}_{\mathbf{x}}$ and $\hat{\Psi}_{\mathbf{u}}$, while Kim et al. [15] proposed to use spherical Gaussian noises, we instead set them as an empirical covariance matrices of the samples collected from the optimization trajectory. Specifically, we take additional $K$ steps from each MAP solution to compute the empirical covariance.

$$
\theta_{\mathbf{x}}^{(0)} = \theta_{\mathbf{x}}, \quad \theta_{\mathbf{x}}^{(t)} = \texttt{opt}(\theta_{\mathbf{x}}^{(t-1)}, (\mathbf{x}, \mathbf{y})), \quad \hat{\mu}_{\mathbf{x}} = g_{\texttt{sg}(\theta_{\mathbf{x}}^{(0)})}(\mathbf{u}),
$$

$$
\hat{\Psi}_{\mathbf{x}} := \texttt{sg}\left( \mathrm{diag}\left( \frac{1}{K}\sum_{k=1}^{K} g_{\theta_{\mathbf{x}}^{(k)}}^2(\mathbf{u}) - \left( \frac{1}{K}\sum_{k=1}^{K} g_{\theta_{\mathbf{x}}^{(k)}}(\mathbf{u}) \right)^2 \right) \right),
\tag{16}
$$

where $\texttt{opt}(\theta, \mathbf{x})$ is a step of SGD optimization applied to $\theta$ with data $\mathbf{x}$ and the squares in the $\mathrm{diag}(\cdot)$ are applied in element-wise manner. Note also that we are applying the stop-gradient operations for to block the gradient flow that might lead to complication in the backpropagation procedure. The variational parameters $(\hat{\mu}_{\mathbf{u}}, \hat{\Psi}_{\mathbf{u}})$ are constructed in a similar fashion, but using the psedocoreset $(\mathbf{u}, \tilde{\mathbf{y}})$ instead of the original data $(\mathbf{x}, \mathbf{y})$. It is noteworthy that our approach is similar to one of the methods in Bayesian learning, SWAG [18]. However, while SWAG focuses on collecting statistics on weight space trajectories, our method constructs statistics in function spaces. This distinction makes our approach more suitable and scalable for pseudocoreset construction. The overview of proposed method is provided in Fig. 1.

With the variational approximations constructed as described, we obtain a Monte-Carlo estimator of Eq. 14,

$$
\begin{aligned}
\nabla_{\mathbf{u}} D_{\mathrm{KL}}[\nu_{\mathbf{x}} | \nu_{\mathbf{u}}] &\approx -\nabla_{\mathbf{u}} \mathbb{E}_{q_{\mathbf{x}}(\mathbf{f}_{\mathbf{u}})}[\log p(\tilde{\mathbf{y}} \,|\, \mathbf{f}_{\mathbf{u}})] + \mathbb{E}_{q_{\mathbf{u}}(\mathbf{f}_{\mathbf{u}})}\left[ \nabla_{\mathbf{u}} \log p(\tilde{\mathbf{y}} \,|\, \mathbf{f}_{\mathbf{u}}) \right] \\
&= -\nabla_{\mathbf{u}} \mathbb{E}_{p(\varepsilon_{\mathbf{x}})}[\log p(\tilde{\mathbf{y}} \,|\, \hat{\mu}_{\mathbf{x}} + \hat{\Psi}_{\mathbf{x}}^{1/2}\varepsilon_{\mathbf{x}})] + \mathbb{E}_{p(\varepsilon_{\mathbf{u}})}\left[ \nabla_{\mathbf{u}} \log p(\tilde{\mathbf{y}} \,|\, \hat{\mu}_{\mathbf{u}} + \hat{\Psi}_{\mathbf{u}}^{1/2}\varepsilon_{\mathbf{u}}) \right] \\
&\approx \frac{1}{S}\sum_{s=1}^{S}\left( -\nabla_{\mathbf{u}} \log p\left(\tilde{\mathbf{y}} \,|\, \hat{\mu}_{\mathbf{x}} + \hat{\Psi}_{\mathbf{x}}^{1/2}\varepsilon_{\mathbf{x}}^{(s)}\right) + \nabla_{\mathbf{u}} \log p\left(\tilde{\mathbf{y}} \,|\, \hat{\mu}_{\mathbf{u}} + \hat{\Psi}_{\mathbf{u}}^{1/2}\varepsilon_{\mathbf{u}}^{(s)}\right) \right),
\end{aligned}
\tag{17}
$$

where $p(\varepsilon_{\mathbf{x}})$ and $p(\varepsilon_{\mathbf{u}})$ are standard Gaussians, $(\varepsilon_{\mathbf{x}}^{(s)})_{s=1}^{S}$ and $(\varepsilon_{\mathbf{u}}^{(s)})_{s=1}^{S}$ are i.i.d. samples from them.

---

**Algorithm 1** Multi-architecture Function space Bayesian Pseudocoreset

---

**Require:** Set of architectures $\mathcal{A}$, expert trajectories $\{\mathcal{E}^{(a)} : a \in \mathcal{A}\}$, prior distributions of parameters $\{\pi_0^{(a)} :$
   $a \in \mathcal{A}\}$, an optimizer `opt`.
   Initialize $\mathbf{u}$ with random minibatch of coreset size $m$.
   **for** $i = 1, \ldots, N$ **do**
       Initialize the gradient of pseudocoreset, $\mathbf{g} \leftarrow 0$.
       **for** $a \in \mathcal{A}$ **do**
           Sample the MAP solution computed for $\mathbf{x}$, $\theta_{\mathbf{x}} \in \mathcal{E}^{(a)}$.
           Sample an initial random parameter $\theta_0 \sim \pi_0^{(a)}(\theta)$.
           **repeat**
               $\theta_t \leftarrow \text{opt}(\theta_{t-1}, (\mathbf{u}, \tilde{\mathbf{y}}))$
           **until** converges to obtain the MAP solution computed for $\mathbf{u}$, $\theta_{\mathbf{u}}$.
           Obtain $\hat{\mu}_{\mathbf{x}}, \hat{\mu}_{\mathbf{u}}, \hat{\Psi}_{\mathbf{x}}, \hat{\Psi}_{\mathbf{u}}$ by Eq. 16.
           Compute the pseudocoreset gradient $\mathbf{g}^{(a)}$ using Eq. 17.
           $\mathbf{g} \leftarrow \mathbf{g} + \mathbf{g}^{(a)}$.
       **end for**
       Update the pseudocoreset $\mathbf{u}$ by using the gradient $\mathbf{g}$.
   **end for**

---

### 3.4 Multiple architectures FBPC training

One significant advantage of function space posterior matching is that the function is typically of much lower dimension compared to the weight. This makes it more likely for function spaces to exhibit similar posterior shapes in the vicinity of the MAP solutions. This characteristic of function space encourages the exploration of function space pseudocoreset training in the context of multiple architectures. Because, the task of training a coreset that matches the highly complex weight space posterior across multiple architectures is indeed challenging, while the situation becomes relatively easier when dealing with architectures that exhibit similar function posteriors.

Therefore we propose a novel multi-architecture FBPC algorithm in Algorithm 1. The training procedure involves calculating the FBPC losses for each individual architecture separately and then summing them together to update. This approach allows us to efficiently update the pseudocoreset by considering the contributions of each architecture simultaneously. We will empirically demonstrate that this methodology significantly enhances the architecture generalization ability of pseudocoresets in Section 5.

### 3.5 Compare to weight space Bayesian pseudocoresets

By working directly on the function space, our method could bypass several challenges that may arise when working on a weight space. Indeed, a legitimate concern arises regarding multi-modality, as the posterior distributions of deep neural networks are highly complex. It makes the optimization of pseudocoresets on weight space difficult. Moreover, minimization of weight space divergence does not necessarily guarantee proximity in the function space. Consequently, although we try to minimize the weight space divergence, there is a possibility that the obtained function posterior may significantly deviate from the true posterior. However, if we directly minimize the divergence between the function distributions, we can effectively address this issue.

On the other hand, there is an additional concern related to memory limitations. While it has been demonstrated in Kim et al. [15] that the memory usage of Bayesian pseudocoresets employing forward KL divergence is not excessively high, we can see that Eq. 7 requires Monte-Carlo samples of weights, which requires $\mathcal{O}(Sp)$ where $S$ and $p$ represent the number of Monte-Carlo samples and the dimensionality of the weights, respectively. This dependence on Monte-Carlo sampling poses a limitation for large-scale networks when memory resources are constrained. In contrast, our proposed method requires significantly less memory, $\mathcal{O}(Sd)$ where $d$ represents the dimensionality of the functions. Indeed, all the results presented in this paper were obtained using a single NVIDIA RTX-3090 GPU with 24GB VRAM.

# 4    Related work

**Bayesian coresets**    Bayesian Coreset [3–5, 14] is a field of research aimed at addressing the computational challenges of MCMC and VI on large datasets in terms of time and space complexity [9, 1, 26]. It aims to approximate the energy function of the entire dataset using a weighted sum of a small subset. However for high-dimensional data, Manousakas et al. [19] demonstrates that considering only subsets as Bayesian coreset is not sufficient, as the KL divergence between the approximated coreset posterior and the true posterior increases with the data dimension, and they proposed Bayesian pseudocoresets. There are recent works on constructing pseudocoreset variational posterior to be more flexible [8] or how to effectively optimize the divergences between posteriors [15, 8, 20, 21]. However, there is still a limitation in constructing high-dimensional Bayesian pseudocoresets specifically for deep neural networks.

**Dataset distillation**    Dataset distillation also aims to synthesize the compact datasets that capture the essence of the original dataset. However, the dataset distillation places particulary on optimizing the test performance of the distilled dataset. Consequently, the primary objective in dataset distillation is to maximize the performance of models trained using the distilled dataset, and researchers provides how to effectively solve this bi-level optimization [33, 23, 22, 40, 36, 39, 7]. In recent work, Kim et al. [15] established a link between specific dataset distillation methods and optimizing certain divergence measures associated with Bayesian pseudocoresets.

**Function space variational inference**    Although Bayesian neural networks exhibit strong capabilities in performing variational inference, defining meaningful priors or efficiently inferring the posterior on weight space is still challenging due to their over-parametrization. To overcome this issue, researchers have increasingly focused on function space variational inference [6, 2, 16, 30, 25, 32, 27, 17]. For instance, Sun et al. [31] introduced a framework that formulates the KL divergence between functions as the supremum of marginal KL divergences over finite sets of inputs. Wang et al. [34] utilizes particle-based optimization directly in the function space. Furthermore, Rudner et al. [28] recently proposed a scalable method for function space variational inference on deep neural networks.

# 5    Experiments

## 5.1    Experimental Setup

In our study, we employed the CIFAR10, CIFAR100 and Tiny-ImageNet datasets to create Bayesian pseudocoresets of coreset size $m \in \{1, 10, 50\}$ images per class (ipc). These pseudocoresets were then evalutated by conducting the Stochastic Gradient Hamiltonian Monte Carlo (SGHMC) [9] algorithm on those pseudocoresets. We measured the top-1 accuracy and negative log-likelihood of the SGHMC algorithm on the respective test datasets. Following the experimental setup of previous works [15, 7, 39], we use the differentiable siamese augmentation [37]. For the network architectures, we use 3-layer ConvNet for CIFAR10 and CIFAR100, and 4-layer ConvNet for Tiny-ImageNet.

We employed three baseline methods to compare the effectiveness of function space Bayesian pseudocoresets. The first baseline is the random coresets, which involves selecting a random mini-batch of the coreset size. The others two baseline methods, BPC-rKL [15, 19] and BPC-fKL [15], are Bayesian pseudocoresets on weight space. BPC-rKL and BPC-fKL employ reverse KL divergence and forward KL divergence as the divergence measures for their training, respectively.

## 5.2    Main Results

Table 1 and Table 2 show the results of each baseline and our method for each dataset. For BPC-rKL and BPC-fKL, we used the official code from [15] for training the pseudocoresets, and only difference is that we used our own SGHMC hyperparameters during evaluation. For detailed experiment setting, please refer to Appendix C.

As discussed earlier, we utilized the empirical covariance to variational posterior instead of using naïve isotropic Gaussian for the function space variational posterior. To assess the effectiveness of using sample covariance, we compare these two in Table 1, as we also presented the results for

**Table 1:** Averaged test accuracy and negative log-likelihoods of models trained on each Bayesian pseudocoreset from scratch using SGHMC on the CIFAR10 dataset. Bold is the best and underline is the second best. These values are averaged over 5 random seeds.

| ipc | SGHMC | Random | BPC-rKL [15, 19] | BPC-fKL [15] | FBPC-isotropic (Ours) | FBPC (Ours) |
|---|---|---|---|---|---|---|
| 1 | Acc (↑) | $16.30_{\pm 0.74}$ | $20.44_{\pm 1.06}$ | $\underline{34.50}_{\pm 1.62}$ | $32.00_{\pm 0.75}$ | $\mathbf{35.45}_{\pm 0.31}$ |
|  | NLL (↓) | $4.66_{\pm 0.03}$ | $4.51_{\pm 0.10}$ | $3.86_{\pm 0.13}$ | $\mathbf{3.40}_{\pm 0.27}$ | $\underline{3.79}_{\pm 0.04}$ |
| 10 | Acc (↑) | $32.48_{\pm 0.34}$ | $37.92_{\pm 0.66}$ | $56.19_{\pm 0.61}$ | $\underline{61.43}_{\pm 0.35}$ | $\mathbf{62.33}_{\pm 0.34}$ |
|  | NLL (↓) | $2.98_{\pm 0.03}$ | $2.47_{\pm 0.04}$ | $1.48_{\pm 0.02}$ | $\underline{1.35}_{\pm 0.02}$ | $\mathbf{1.31}_{\pm 0.02}$ |
| 50 | Acc (↑) | $49.68_{\pm 0.46}$ | $51.86_{\pm 0.38}$ | $64.74_{\pm 0.32}$ | $\mathbf{71.33}_{\pm 0.19}$ | $\underline{71.23}_{\pm 0.17}$ |
|  | NLL (↓) | $2.06_{\pm 0.02}$ | $1.95_{\pm 0.02}$ | $\underline{1.26}_{\pm 0.01}$ | $\mathbf{1.03}_{\pm 0.01}$ | $\mathbf{1.03}_{\pm 0.05}$ |

**Table 2:** Averaged test accuracy and negative log-likelihoods of models trained on each Bayesian pseudocoreset from scratch using SGHMC on the CIFAR100 and Tiny-ImageNet datasets. These values are averaged over 3 random seeds.

| | | CIFAR100 | | | Tiny-ImageNet | | |
|---|---|---|---|---|---|---|---|
| | ipc | 1 | 10 | 50 | 1 | 10 | 50 |
| Random | Acc (↑) | $4.82_{\pm 0.47}$ | $18.0_{\pm 0.31}$ | $35.1_{\pm 0.23}$ | $1.90_{\pm 0.08}$ | $7.21_{\pm 0.04}$ | $19.15_{\pm 0.12}$ |
| | NLL (↓) | $5.55_{\pm 0.07}$ | $4.57_{\pm 0.01}$ | $3.35_{\pm 0.01}$ | $6.18_{\pm 0.04}$ | $5.77_{\pm 0.02}$ | $4.88_{\pm 0.01}$ |
| BPC-fKL | Acc (↑) | $14.7_{\pm 0.16}$ | $28.1_{\pm 0.60}$ | $37.1_{\pm 0.33}$ | $3.98_{\pm 0.13}$ | $11.4_{\pm 0.45}$ | $22.18_{\pm 0.32}$ |
| | NLL (↓) | $4.17_{\pm 0.05}$ | $3.53_{\pm 0.05}$ | $3.28_{\pm 0.24}$ | $5.63_{\pm 0.03}$ | $5.08_{\pm 0.05}$ | $4.65_{\pm 0.02}$ |
| FBPC (Ours) | Acc (↑) | $\mathbf{21.0}_{\pm 0.76}$ | $\mathbf{39.7}_{\pm 0.31}$ | $\mathbf{44.47}_{\pm 0.35}$ | $\mathbf{10.14}_{\pm 0.68}$ | $\mathbf{19.42}_{\pm 0.51}$ | $\mathbf{26.43}_{\pm 0.31}$ |
| | NLL (↓) | $\mathbf{3.76}_{\pm 0.11}$ | $\mathbf{2.67}_{\pm 0.02}$ | $\mathbf{2.63}_{\pm 0.01}$ | $\mathbf{4.69}_{\pm 0.05}$ | $\mathbf{4.14}_{\pm 0.02}$ | $\mathbf{4.30}_{\pm 0.05}$ |

FBPC-isotropic, which represents the FBPC trained with a unit covariance Gaussian posteriors. The results clearly demonstrate that using sample covariance captures valuable information from the posterior distribution, resulting in improved performance. Overall, the results presented in Table 1 and Table 2 demonstrate that our method also outperforms the baseline approaches, including random coresets, BPC-rKL and BPC-fKL, in terms of both accuracy and negative log-likelihood, especially on the large-scale datasets in Table 2.

Furthermore, the Bayesian pseudocoreset can be leveraged to enhance robustness against distributional shifts when combined with Bayesian model averaging. To assess the robustness of our function space Bayesian pseudocoresets on out-of-distribution inputs, we also conducted experiments using the CIFAR10-C dataset [11]. This dataset involves the insertion of image corruptions into the CIFAR10 images. By evaluating the performance of the pseudocoresets on CIFAR10-C, we can see the model's ability to handle input data that deviates from the original distribution. In Table 3, we provide the results for top-1 accuracy and degradation scores, which indicate the extent to which accuracy is reduced compared to the in-distribution's test accuracy. The result demonstrates that our FBPC consistently outperforms the weight space Bayesian pseudocoreset, BPC-fKL.

### 5.3 Architecture generalization

In this section, we aim to demonstrate the architecture generalizability of FBPC and emphasize the utility of multi-architecture training as we discussed in the previous section. We specifically focus on investigating the impact of varying normalization layers on the generalizability of the pseudocoreset, since it is widely recognized that a pseudocoreset trained using one architecture may struggle to generalize effectively to a model that employs different normalization layers. We have also included the results of cross-architecture experiments that involve changing the architecture itself in Appendix D.1.

To assess this, we compare a single architecture trained pseudocoreset and a multiple architecture trained pseudocoreset. For single architecture training, we initially train a pseudocoreset using one architecture with a specific normalization layer, for instance we use instance normalization. Subsequently, we evaluate the performance of this pseudocoreset on four different types of normalization layers: instance normalization, group normalization, layer normalization, and batch normalization. For multiple architecture training, we aggregate four losses for single architecture training of each architecture, and train the pseudocoreset with the sum of all losses, as mentioned in previous section.

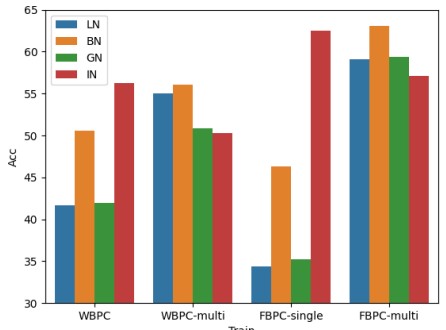

(a) Performance evaluation with different normalization layers. Color represents the test architectures. The multiple architecture FBPC training enhances the generalization ability to other architectures.

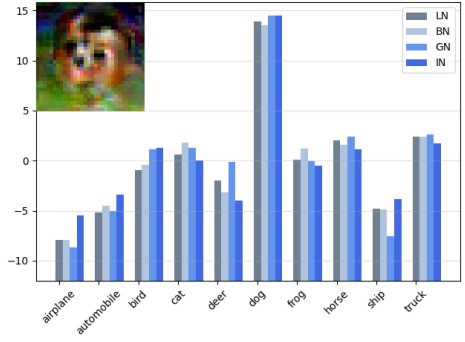

(b) A sample image and its corresponding function values. Despite variations in the normalization layers of different architectures, the function values exhibit similarity across the architectures.

**Figure 2:** Results for multiple architecture FBPC training.

**Table 3:** Test accuracy and degradation scores of models trained on each Bayesian pseudocoreset from scratch using SGHMC on the CIFAR10-C. Degradation refers to the extent by which a model's accuracy decreases when evaluated on the CIFAR10-C dataset compared to the CIFAR10 test dataset.

| | corruption | BN | DB | ET | FT | GB | JPEG | MB | PIX | SN | SP | Avg. |
|---|---|---|---|---|---|---|---|---|---|---|---|---|
| BPC-fKL | Acc ($\uparrow$) | 33.5 | 34 | 35.9 | 25.1 | 33.7 | 39.1 | 32.7 | 38.3 | 28.9 | 41.2 | 34.3 |
| | Degradation ($\downarrow$) | 40.3 | 39.4 | 36 | 55.2 | 39.9 | 30.3 | 41.6 | 31.6 | 48.4 | 26.5 | 38.9 |
| FBPC | Acc ($\uparrow$) | 48.9 | 46.4 | 47.6 | 41.7 | 44.0 | 52.0 | 44.3 | 51.0 | 47.1 | 52.3 | **47.5** |
| | Degradation ($\downarrow$) | 21.5 | 25.7 | 23.7 | 33.0 | 29.3 | 16.4 | 28.8 | 18.1 | 24.4 | 16.1 | **23.7** |

As depicted in Fig. 2a, we observe that both WBPC (Weight space Bayesian pseudocoresets) and FBPC-single (Function space Bayesian pseudocoresets trained on a single architecture) exhibit a notable trend, that they tend to not perform well when evaluated on the architecture that incorporates different normalizations, regardless of whether it is trained on weight space or function space. On the other hand, when trained with multiple architectures, both WBPC-multi and FBPC-multi perform well across the all architectures, while notably FBPC-multi significantly outperforms WBPC-multi.

As mentioned in the previous section, we hypothesize that the superior performance of FBPC compared to WBPC can be attributed to the likelihood of having similar function space posterior across architectures. To validate this, we conduct an examination of the logit values for each sample across different architectures. As an illustration, we provide an example pseudocoreset image belonging to the class label "dog" along with its corresponding logits for all four architectures. As Fig. 2b shows, it can be observed that the logits display a high degree of similarity, indicating a strong likelihood of matching function posterior distributions. Our analysis confirms that, despite architectural disparities, the function spaces generated by these architectures exhibit significant similarity and it contributes to superiority of FBPC in terms of architecture generalizability.

## 6 Conclusion

In this paper, we explored the function space Bayesian pseudocoreset. We constructed it by minimizing forward KL divergence between the function posteriors of pseudocoreset and the entire dataset. To optimize the divergence, we proposed a novel method to effectively approximate the function posteriors with an efficient training procedure. Finally, we empirically demonstrated the superiority of our function space Bayesian pseudocoresets compared to weight space Bayesian pseudocoresets, in terms of test performance, uncertainty quatification, OOD robustness, and architectural robustness.

**Limitation** Despite showing promising results on function space Bayesian pseudocoresets, there still exist a few limitations in the training procedure. Our posterior approximation strategy requires

the MAP solutions, which necessitates training them prior to each update step or preparing expert trajectories. This can be time-consuming and requires additional memory to store the expert trajectories.

**Societal Impacts** Our work is hardly likely to bring any negative societal impacts.

## Acknowledgments

This work was supported by KAIST-NAVER Hypercreative AI Center, Institute of Information & communications Technology Planning & Evaluation (IITP) grant funded by the Korea government (MSIT) (No.2019-0-00075, Artificial Intelligence Graduate School Program (KAIST), No.2022-0-00184, Development and Study of AI Technologies to Inexpensively Conform to Evolving Policy on Ethics, and No.2022-0-00713, Meta-learning Applicable to Real-world Problems) and the National Research Foundation of Korea (NRF) grant funded by the Korea government (MSIT) (No. 2022R1A5A708390812)

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

# A  Proofs

**Proposition A.1.** *The gradient of the forward KL divergence with respect to the coreset $\mathbf{u}$ is*
$$\nabla_{\mathbf{u}} D_{\mathrm{KL}}[\nu_{\mathbf{x}} \| \nu_{\mathbf{u}}] = -\nabla_{\mathbf{u}} \mathbb{E}_{[\nu_{\mathbf{x}}]_{\mathbf{u}}}[\log p(\tilde{\mathbf{y}} \,|\, \mathbf{f}_{\mathbf{u}})] + \mathbb{E}_{[\nu_{\mathbf{u}}]_{\mathbf{u}}}[\nabla_{\mathbf{u}} \log p(\tilde{\mathbf{y}} \,|\, \mathbf{f}_{\mathbf{u}})], \tag{10}$$
*where $[\nu_{\mathbf{x}}]_{\mathbf{u}}$ and $[\nu_{\mathbf{u}}]_{\mathbf{u}}$ are finite-dimensional distributions of the stochastic processes $\nu_{\mathbf{x}}$ and $\nu_{\mathbf{u}}$, respectively, $\mathbf{f}_{\mathbf{u}} := (f(u_j))_{j=1}^{m}$, and $p(\tilde{\mathbf{y}} \,|\, \mathbf{f}_{\mathbf{u}}) = \prod_{j=1}^{m} p(\tilde{y}_j \,|\, f(u_j))$.*

*Proof.* We follow the arguments in de G. Matthews et al. [10], Rudner et al. [29]. The forward KL divergence is defined as,
$$D_{\mathrm{KL}}[\nu_{\mathbf{x}} \| \nu_{\mathbf{u}}] = \int \log \frac{\mathrm{d}\nu_{\mathbf{x}}}{\mathrm{d}\nu_{\mathbf{u}}}(f) \mathrm{d}\nu_{\mathbf{x}}(f). \tag{18}$$
By the chain rule for the Radon-Nikodym derivative, we have
$$D_{\mathrm{KL}}[\nu_{\mathbf{x}} \| \nu_{\mathbf{u}}] = \int \log \frac{\mathrm{d}\nu_{\mathbf{x}}}{\mathrm{d}\nu_0}(f) \mathrm{d}\nu_{\mathbf{x}}(f) - \int \log \frac{\mathrm{d}\nu_{\mathbf{u}}}{\mathrm{d}\nu_0}(f) \mathrm{d}\nu_{\mathbf{x}}(f). \tag{19}$$
The first term does not depend on $\mathbf{u}$, so we investigate the second term. By the measure theoretic Bayes' rule,
$$\frac{\mathrm{d}\nu_{\mathbf{u}}}{\mathrm{d}\nu_0}(f) = \frac{p(\tilde{\mathbf{y}} \,|\, f, \mathbf{u})}{p(\tilde{\mathbf{y}} \,|\, \mathbf{u})}, \tag{20}$$
where $p(\tilde{\mathbf{y}} \,|\, f, \mathbf{u}) := \prod_{j=1}^{m} p(\tilde{y}_j \,|\, u_j, f)$ and,
$$p(\tilde{\mathbf{y}}|\mathbf{u}) = \int p(\tilde{\mathbf{y}}|f, \mathbf{u}) \mathrm{d}\nu_0(f). \tag{21}$$
Now let $\rho_A : (\mathcal{X} \to \mathbb{R}^d) \to (A \to \mathbb{R}^d)$ be a projection function that takes a function $f$ and returns its restriction on a set $A \subseteq \mathcal{X}$. Assuming that the likelihood depends only on the finite index set $\mathbf{u}$, we can write
$$\frac{\mathrm{d}\nu_{\mathbf{u}}}{\mathrm{d}\nu_0}(f) = \frac{\mathrm{d}[\nu_{\mathbf{u}}]_{\mathbf{u}}}{\mathrm{d}[\nu_0]_{\mathbf{u}}}(\rho_{\mathbf{u}}(f)) = \frac{p(\tilde{\mathbf{y}} \,|\, \mathbf{f}_{\mathbf{u}})}{p(\tilde{\mathbf{y}} \,|\, \mathbf{u})}, \tag{22}$$
where $[\cdot]_{\mathbf{u}}$ denotes the finite-dimensional distribution of stochastic process evaluated at $\mathbf{u}$ and $\rho_{\mathbf{u}}(f) := \mathbf{f}_{\mathbf{u}} := (f(u_j))_{j=1}^{m}$ are corresponding function values at $\mathbf{u}$. Putting this back into the above equation,
$$\begin{aligned}
\int \log \frac{\mathrm{d}\nu_{\mathbf{u}}}{\mathrm{d}\nu_0}(f) \mathrm{d}\nu_{\mathbf{x}}(f) &= \int \log \frac{\mathrm{d}[\nu_{\mathbf{u}}]_{\mathbf{u}}}{\mathrm{d}[\nu_0]_{\mathbf{u}}}(\mathbf{f}_{\mathbf{u}}) \mathrm{d}[\nu_{\mathbf{x}}]_{\mathbf{u}}(\mathbf{f}_{\mathbf{u}}) \\
&= \int \log \frac{p(\tilde{\mathbf{y}} \,|\, \mathbf{f}_{\mathbf{u}})}{p(\tilde{\mathbf{y}} \,|\, \mathbf{u})} \mathrm{d}[\nu_{\mathbf{x}}]_{\mathbf{u}}(\mathbf{f}_{\mathbf{u}}) \\
&= \mathbb{E}_{[\nu_{\mathbf{x}}]_{\mathbf{u}}}[\log p(\tilde{\mathbf{y}} \,|\, \mathbf{f}_{\mathbf{u}})] - \log p(\tilde{\mathbf{y}} \,|\, \mathbf{u}).
\end{aligned} \tag{23}$$
Now taking the gradient w.r.t. $\mathbf{u}$, we get
$$\nabla_{\mathbf{u}} D_{\mathrm{KL}}[\nu_{\mathbf{x}} \| \nu_{\mathbf{u}}] = -\nabla_{\mathbf{u}} \mathbb{E}_{[\nu_{\mathbf{x}}]_{\mathbf{u}}}[\log p(\tilde{\mathbf{y}} \,|\, \mathbf{f}_{\mathbf{u}})] + \nabla_{\mathbf{u}} \log p(\tilde{\mathbf{y}} \,|\, \mathbf{u}). \tag{24}$$
Note also that
$$\begin{aligned}
\nabla_{\mathbf{u}} \log p(\tilde{\mathbf{y}} \,|\, \mathbf{u}) &= \nabla_{\mathbf{u}} \log \int p(\tilde{\mathbf{y}} \,|\, f, \mathbf{u}) \mathrm{d}\nu_0(f) \\
&= \int \frac{\nabla_{\mathbf{u}} p(\tilde{\mathbf{y}} \,|\, f, \mathbf{u})}{p(\tilde{\mathbf{y}} \,|\, \mathbf{u})} \mathrm{d}\nu_0(f) \\
&= \int \nabla_{\mathbf{u}} \log p(\tilde{\mathbf{y}} \,|\, f, \mathbf{u}) \frac{p(\tilde{\mathbf{y}} \,|\, f, \mathbf{u})}{p(\tilde{\mathbf{y}} \,|\, \mathbf{u})} \mathrm{d}\nu_0(f) \\
&= \int \nabla_{\mathbf{u}} \log p(\tilde{\mathbf{y}} \,|\, f, \mathbf{u}) \frac{\mathrm{d}\nu_{\mathbf{u}}}{\mathrm{d}\nu_0}(f) \mathrm{d}\nu_0(f) \\
&= \int \nabla_{\mathbf{u}} \log p(\tilde{\mathbf{y}} \,|\, f, \mathbf{u}) \mathrm{d}\nu_{\mathbf{u}}(f) \\
&= \int \nabla_{\mathbf{u}} \log p(\tilde{\mathbf{y}} \,|\, \mathbf{f}_{\mathbf{u}}) \mathrm{d}[\nu_{\mathbf{u}}]_{\mathbf{u}}(f) = \mathbb{E}_{[\nu_{\mathbf{u}}]_{\mathbf{u}}}[\nabla_{\mathbf{u}} \log p(\tilde{\mathbf{y}} \,|\, \mathbf{f}_{\mathbf{u}})].
\end{aligned} \tag{25}$$
As a result, we conclude that
$$\nabla_{\mathbf{u}} D_{\mathrm{KL}}[\nu_{\mathbf{x}} \| \nu_{\mathbf{u}}] = -\nabla_{\mathbf{u}} \mathbb{E}_{[\nu_{\mathbf{x}}]_{\mathbf{u}}}[\log p(\tilde{\mathbf{y}} \,|\, \mathbf{f}_{\mathbf{u}})] + \mathbb{E}_{[\nu_{\mathbf{u}}]_{\mathbf{u}}}[\nabla_{\mathbf{u}} \log p(\tilde{\mathbf{y}} \,|\, \mathbf{f}_{\mathbf{u}})]. \tag{26}$$
$\square$

# B  Inducing points in Stochastic Variational Gaussian Processes

Stochastic Variational Gaussian Processes [SVGP; 12, 13] were introduced as a solution to address the significant computational complexity, characterized by a cubic computational complexity of $O(n^3)$ and memory cost of $O(n^2)$, associated with performing inference using Gaussian Processes. In this context, $n$ represents the total number of training data points. SVGP effectively leverages a concept called *inducing points*, which serves to reduce the computational complexity to $O(m^3)$ and memory requirements to $O(m^2)$ during inference, while still providing a reliable approximation of the posterior distribution of the entire training dataset. Notably, $m$ denotes the number of inducing points, which is typically much smaller than the total number of training data points i.e. $m \ll n$. The above description clearly shows that the inducing points have a similar purpose to FBPC. However, there are some differences in their learning objectives. In the context of SVGP, the process of optimizing the inducing points denoted as $Z = \{z_1, \ldots, z_m\}$ involves maximizing the ELBO in order to make a variational Gaussian distribution $q(\mathbf{f}_{\mathrm{tr}}, \mathbf{f}_z)$ well approximate the posterior distribution $p(\mathbf{f}_{\mathrm{tr}}, \mathbf{f}_z | \mathbf{y}_{\mathrm{tr}})$. This variational distribution is composed of two parts: 1) $p(\mathbf{f}_{\mathrm{tr}} | \mathbf{f}_z)$, which represents the Gaussian posterior distribution, and 2) $q(\mathbf{f}_z)$, which is the Gaussian variational distribution. During this optimization, we focus on training the inducing points $Z$ as well as the mean and variance of the variational distribution $q(\mathbf{f}_z)$. The goal of this optimization is to create a good approximation of the posterior distribution $p(y_{\mathrm{te}} | x_{\mathrm{te}}, D_{\mathrm{tr}})$ during the inference process, all while keeping the computational cost low. On the other hand, as outlined in Section 3.2, the formulation of FBPC involves directly minimizing the divergence between function space posterior, specifically $D_{\mathrm{KL}}[\nu_{\mathbf{x}} \| \nu_{\mathbf{u}}]$. To sum up, while they do share some similarities in that they introduce a set of learnable pseudo data points, they are fundamentally different in their learning objectives. SVGP is interested in approximating the full data posterior through the inducing points, while ours aims to make the pseudocoreset posterior as close as possible to the full data posterior.

# C  Experimetal Details

The code for our experiments will be available soon.

## C.1  Expert trajectory

For expert trajectory, we trained the network with the entire dataset and saved their snapshot parameters at every epoch, following the setup described in [7]. For training, we used an SGD optimizer with a learning rate of 0.01. We saved 100 training trajectories, with each trajectory consisting of 50 epochs.

## C.2  Hyperparmeter setting

**Training**   In our training procedure, we have some hyperparameters. Firstly, we sampled the MAP solution of $\mathbf{x}$, denoted as $\theta_{\mathbf{x}}$, from the later part of the expert trajectories. The decision of how many samples from the later part to utilize was treated as a hyperparameter for each experimental setting. We chose to use samples from $T$ epoch onwards as the MAP solution samples. When obtaining the MAP solution $\theta_{\mathbf{u}}$, there are also several hyperparameters involved, the optimizer and convergence criteria for training the MAP solution from random parameters. We used an Adam optimizer with a learning rate of 0.001 to train the model until the training loss reached a threshold of $\gamma$ or below.

Next, to approximate our Gaussian variational function posterior, we employed the empirical covariance of the function samples. During the process of drawing function samples, we performed an additional training steps. This step involved specifying the optimizer and the number of steps used. We used an SGD optimizer with learning rates of $\lambda_{\mathbf{x}}$ and $\lambda_{\mathbf{u}}$ for a total of 30 steps during the additional training step for drawing function samples for $\mathbf{x}$ and $\mathbf{u}$, respectively.

Lastly, we used the training iteration $N$, a learning rate of $\alpha$ for pseudocoresets, and set the batch size for pseudocoresets to $B$. The hyperparameters used in our paper are summarized in Table 4.

**Evaluation** To implement the SGHMC algorithm, as discussed in [9] and following the recommendations of [9], we employed the SGD with momentum along with an auxiliary noise term.

$$\begin{cases} \Delta\theta = v \\ \Delta v = -\eta\nabla\tilde{U}(x) - \alpha v + \mathcal{N}(0, 2d). \end{cases} \tag{27}$$

we set $\eta = 0.03$, $\alpha = 0.1$, and $d = 0.01/m$, where $m$ represents the coreset size. We perform 1000 epochs of SGHMC and collect samples every 100 epochs.

# D  Additional Experiments

## D.1  Cross-architecture generalization

In order to assess the cross-architecture generalization performance of FBPC, we trained pseudo-corests of sizes {1, 10, 50} using the ConvNet architecture and tested them on various architectures trained from scratch. In addition to the ConvNet architecture used during training, we also evaluated the performance on sophisticated architectures such as ResNet18, ResNet34, VGG, and AlexNet. The results are presented in Table 5. As evident from Table 5, FBPC demonstrates considerable performance even on architectures different from those used during training, highlighting its strong cross-architecture generalization capabilities.

## D.2  Training FBPC on larger neural networks

To evaluate the scalability of our method to large networks, we trained FBPC with the ResNet18 architecture. Training coreset with larger networks, such as ResNet18, has proven to be challenging and has been explored in only a few previous works [38]. This is primarily due to the lack of scalable training methods and the tendency for overfitting when training large networks. As a result, even when evaluating ResNet after training on a smaller network like ConvNet, the performance tends to suffer. Furthermore, it has been reported that coreset training directly on ResNet initially yields lower performance compared to training on ConvNet [40, 35].

Our experiments also revealed a similar trend in our findings as shown in the first column of Table 6. Although FBPC exhibits excellent scalability, making it easy to train on ResNet18 and larger networks, its performance was observed to be lower compared to ConvNet. On the other hand, the second column, ResNet18 + ConvNet, refers to training both ResNet18 and ConvNet simultaneously using the FBPC-multi training approach. In this case, surprisingly, the test performance of ResNet18 actually improved when trained in conjunction with ConvNet using the FBPC-multi training approach. In this case, the ConvNet accuracy was recorded at 60.03, which did not significantly compromise the ConvNet's performance while enhancing the performance of ResNet18. This suggests that training ConvNet acted as a regularizer, preventing overfitting in ResNet18 and enabling it to achieve better performance.

**Table 4:** Hyperparameter for each experiment setting.

|  | ipc | $T$ | $\gamma$ | $\lambda_{\mathbf{x}}$ | $\lambda_{\mathbf{u}}$ | $N$ | $\alpha$ | $B$ |
|---|---|---|---|---|---|---|---|---|
| | 1 | 1 | 0.01 | 0.05 | 0.01 | 1000 | 100 | 10 |
| CIFAR10 | 10 | 2 | 0.1 | 0.05 | 0.01 | 1000 | 1000 | 100 |
| | 50 | 10 | 0.2 | 0.05 | 0.01 | 1000 | 1000 | 500 |
| | 1 | 2 | 0.1 | 0.1 | 0.1 | 5000 | 1000 | 100 |
| CIFAR100 | 10 | 40 | 0.1 | 0.1 | 0.1 | 5000 | 1000 | 1000 |
| | 50 | 20 | 0.2 | 0.01 | 0.01 | 300 | 1000 | 5000 |
| | 1 | 2 | 0.1 | 0.1 | 3 | 5000 | 1000 | 100 |
| Tiny-ImageNet | 10 | 40 | 1 | 0.1 | 3 | 500 | 1000 | 100 |
| | 50 | 40 | 1 | 0.1 | 3 | 500 | 1000 | 100 |

**Table 5:** Averaged test accuracy and negative log-likelihoods for each architecture of the pseudocoreset trained using the ConvNet architecture with CIFAR10 dataset.

|    |           | ConvNet | ResNet18 | ResNet34 | VGG | AlexNet |
|----|-----------|---------|----------|----------|-----|---------|
| 1  | Acc ($\uparrow$) | $35.71_{\pm 0.90}$ | $27.7_{\pm 0.96}$ | $22.46_{\pm 0.12}$ | $26.33_{\pm 0.88}$ | $21.05_{\pm 0.43}$ |
|    | NLL ($\downarrow$) | $3.44_{\pm 0.07}$ | $2.87_{\pm 0.13}$ | $3.06_{\pm 0.02}$ | $5.38_{\pm 0.74}$ | $3.13_{\pm 0.60}$ |
| 10 | Acc ($\uparrow$) | $62.53_{\pm 0.34}$ | $47.51_{\pm 1.73}$ | $35.48_{\pm 1.22}$ | $47.87_{\pm 1.18}$ | $32.27_{\pm 0.78}$ |
|    | NLL ($\downarrow$) | $1.31_{\pm 0.01}$ | $1.82_{\pm 0.02}$ | $2.41_{\pm 0.01}$ | $3.72_{\pm 0.09}$ | $2.96_{\pm 0.04}$ |
| 50 | Acc ($\uparrow$) | $71.20_{\pm 0.36}$ | $62.02_{\pm 1.55}$ | $47.97_{\pm 3.37}$ | $58.24_{\pm 1.63}$ | $52.42_{\pm 1.30}$ |
|    | NLL ($\downarrow$) | $1.03_{\pm 0.00}$ | $1.41_{\pm 0.05}$ | $2.10_{\pm 0.10}$ | $3.03_{\pm 0.26}$ | $2.08_{\pm 0.07}$ |

**Table 6:** Test performance of FBPC (CIFAR10, ipc 10) on ResNet18. FBPC is trained with ResNet18 and ResNet18 + ConvNet (multiple architecture training).

|           | ResNet18 | ResNet18 + ConvNet |
|-----------|----------|--------------------|
| Acc ($\uparrow$) | 50.00 | 54.90 |
| NLL ($\downarrow$) | 1.75 | 1.56 |

### D.3 Computational cost

To compare how scalable our approach is compared to posterior matching in the weight space, we measured GPU memory usage corresponding to the number of parameters. As shown in Table 7 and Table 8, the memory usage for weight space BPC significantly increases as the number of parameters grows, while FBPC operates very efficiently. Additionally, the coreset ipc increases memory usage proportionally to its size. In terms of memory considerations, FBPC excels. However, as shown in Table 9, in terms of time, our method requires slightly more time because more SGD steps are needed to acquire the empirical covariance. However, when using FBPC-isotropic, these steps can be reduced, trading off a slight decrease in performance for time savings.

### D.4 Differentiable siamese augmentation

Table 10 shows the result for BPC-fKL and FBPC without using DSA [37] and without any augmentation during training. Interestingly, for the ipc 1 case in BPC-fKL, performance improved when DSA was not applied. However, in all other cases, it is evident that not using DSA leads to an average performance drop of approximately 4.7%. Moreover, even when training BPC without augmentation, we observe that function space BPC outperforms weight space BPC.

### D.5 Visualization

In this section, we provide visualizations of the pseudocoreset examples for each dataset.

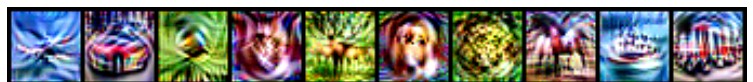

**Figure 3:** Example images of FBPC for CIFAR10 with ipc 1.

**Table 7:** GPU memory usage (GB) for training CIFAR10 FBPC with ipc 10.

|           | LeNet             | ConvNet           | ResNet18          |
|-----------|-------------------|-------------------|-------------------|
| # parameters | $6.2 \times 10^4$ | $3.2 \times 10^5$ | $1.1 \times 10^7$ |
| FBPC      | 0.02              | 0.32              | 2.56              |
| BPC-fKL   | 0.11              | 3.17              | 12.18             |

**Table 8:** GPU memory usage (GB) for training CIFAR10 FBPC according to the ipc.

|         | 1    | 10   | 50    |
|---------|------|------|-------|
| FBPC    | 0.04 | 0.32 | 1.59  |
| BPC-fKL | 0.41 | 3.17 | 15.59 |

**Table 9:** Wall-clock time (sec) for 1 step update for training CIFAR10 pseudocoreset according to the ipc.

|         | 1                | 10               | 50               |
|---------|------------------|------------------|------------------|
| BPC-fKL | $1.04_{\pm 0.10}$ | $1.37_{\pm 0.13}$ | $2.59_{\pm 0.86}$ |
| FBPC    | $1.5_{\pm 0.15}$  | $3.29_{\pm 0.51}$ | $8.38_{\pm 0.48}$ |

**Table 10:** BPC Performances with and without DSA.

|                  | 1                 | 10                | 50                |
|------------------|-------------------|-------------------|-------------------|
| BPC-fKL (no DSA) | $37.26_{\pm 1.65}$ | $50.48_{\pm 1.39}$ | $60.75_{\pm 0.26}$ |
| FBPC (no DSA)    | $33.69_{\pm 2.73}$ | $55.07_{\pm 1.30}$ | $66.03_{\pm 0.21}$ |
| FBPC (DSA)       | $35.45_{\pm 0.31}$ | $62.33_{\pm 0.34}$ | $71.23_{\pm 0.17}$ |

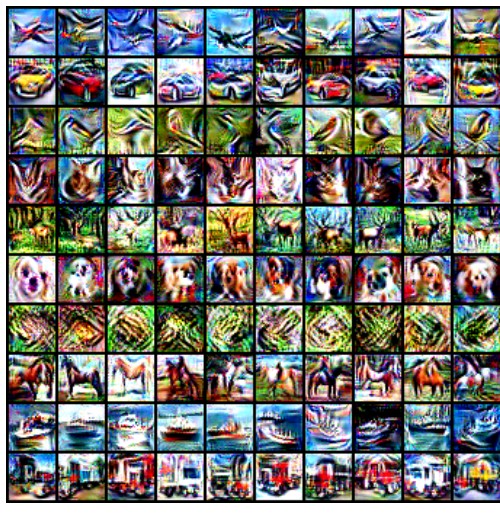 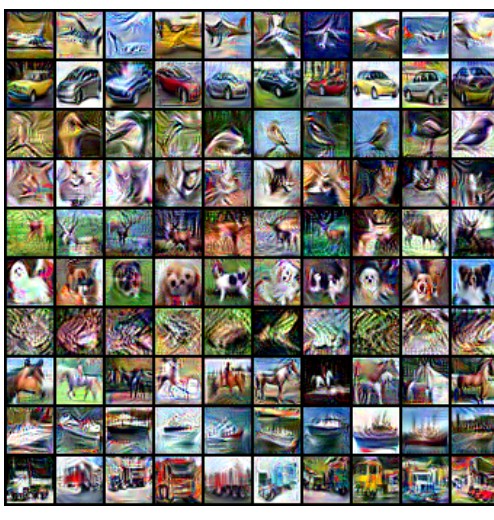

**(a)** Examples images of FBPC for CIFAR10 with ipc 10. 1 image per class.

**(b)** Examples images of FBPC for CIFAR10 with ipc 50. 10 images per class.

**Figure 4:** Examples images of FBPCs for CIFAR10 dataset.

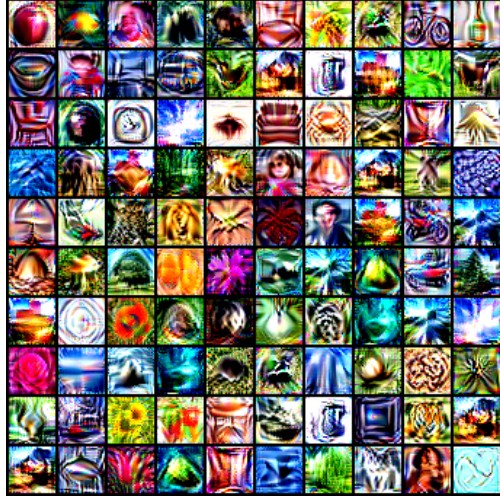

**(a)** Examples images of FBPC for CIFAR100 with ipc 1. 1 image per class.

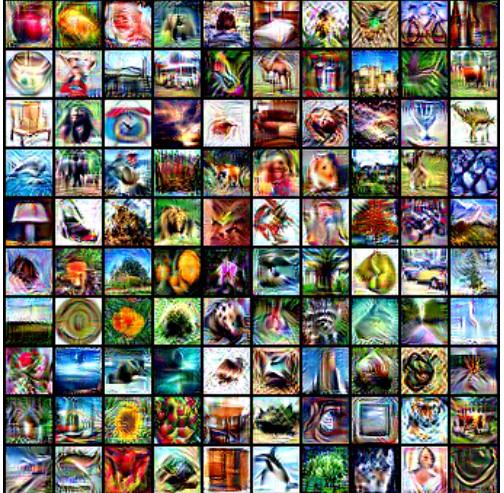

**(b)** Examples images of FBPC for CIFAR100 with ipc 10. 1 image per class.

**Figure 5:** Examples images of FBPCs for CIFAR100 dataset.

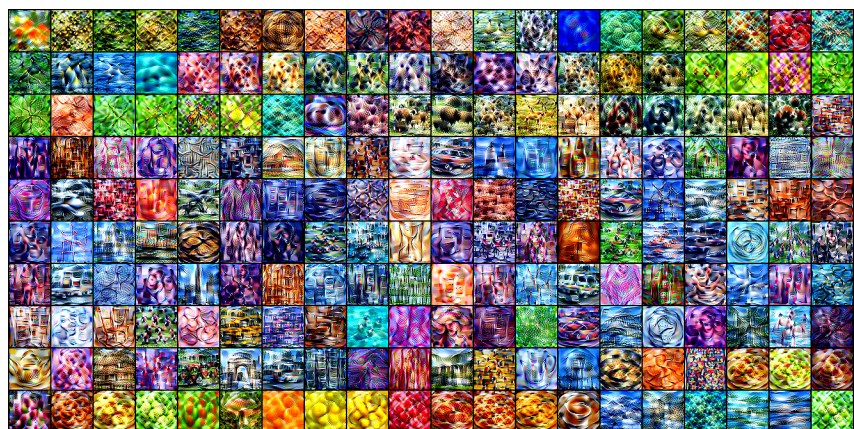

**Figure 6:** Example images of FBPC for Tiny-ImageNet with ipc 1. 1 image per class.

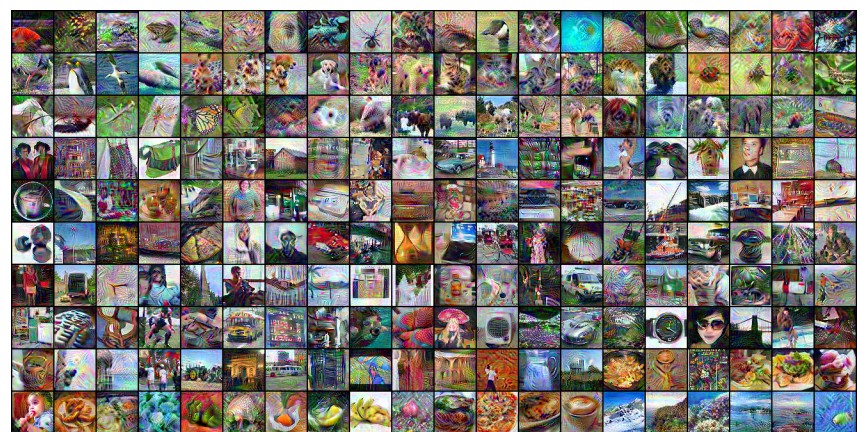

**Figure 7:** Example images of FBPC for Tiny-ImageNet with ipc 10. 1 image per class.

