# OpenReview forum: "Function Space Bayesian Pseudocoreset for Bayesian Neural Networks"
_NeurIPS.cc/2023/Conference — NeurIPS 2023 poster_

### Official Review · Reviewer_jYGC · 2023-07-06

**Soundness:** 4 excellent
**Presentation:** 3 good
**Contribution:** 3 good
**Rating:** 7
**Confidence:** 4

**Summary:**

This paper presents an alternative approach to the construction of Bayesian pseudocoresets by considering the quality of function space approximations. Specifically, they seek to minimise the KL-divergence between function space approximations of posteriors conditioned on the Bayesian pseudocoreset, and the true posterior.

**Strengths:**

* I consider the method very well motivated---the use of variational approximations in weight-space is known to result in pathologies, particularly with high-dimensional models such as BNNs. Function space approximations offer a solution to this by instead computing variational approximations to distributions that exhibit much less multi-modality and related behaviour that is difficult to approximate.
* Although the authors largely follow the work of Rudner et al., they depart from this method in estimating the parameter values of the function space approximation. Instead, they empirical estimates which, although inexact, I expect result in considerable speed-up in computation time.
* Experimental results are good, with the method outperforming the benchmarks.

**Weaknesses:**

* A lot of approximations are involved in the method, and it would be nice to see some evaluation of the effects of each.

**Questions:**

N/A.

**Limitations:**

N/A.

---

> ### Author Rebuttal · Authors · 2023-08-10
>
> We sincerely appreciate your constructive comments. For the weaknesses, please refer to our general responses [G3, G4, G5].

---

> > ### Comment · Reviewer_jYGC · 2023-08-16
> >
> > Many thanks for your response.
> >
> > Regarding [G4]. The Jacobian approximation is not the only approximation used---indeed, the use of the Jacobian is an approximation in itself. A comment on the implications of the linearised Laplace approximation would be greatly appreciated in an updated version of the paper.
> >
> > [G5] discusses the aforementioned points, however doesn't extend beyond comments on scalability of the algorithm. Again, it would be beneficial to the reader to understand the implications of a Gaussian approximation in function space.
> >
> > Nonetheless, I maintain my original score.

---

### Official Review · Reviewer_d2wU · 2023-07-06

**Soundness:** 3 good
**Presentation:** 3 good
**Contribution:** 3 good
**Rating:** 6
**Confidence:** 4

**Summary:**

The following work proposes to a new way to construct Bayesian Pseudo-Coresets. Particularly, the authors propose to optimise the KL-Divergence between posteriors associated with real data and synthetic  in function space rather than the parameter space of large networks. The main argument posed by the authors is that function space is typically lower in dimension as compared to the parameter space of the network. Further, the authors use the theory laid out in [1] to approximate the function space posteriors associated with real data and pseudo-coreset. Further, since the function space can be same for several architectures, the proposed method can also be used with multiple architectures.


**Strengths:**

1.	The argument behind the use of function space seems logical enough, inference in high dimensional parameter space is indeed difficult.
2.	The demonstrated result shows decent improvement compared to SoTA BPC methods.
3.	The “multi-architecture” setting that comes for free because of working in the function space is very intriguing. To the best of my knowledge, this is the first work that enables one to use multiple architectures.

**Weaknesses:**

1. Ambiguous writing.

2. Experiments can be better.

**Questions:**

1.	It seems that the paper could have been better written and presented. Especially for someone who is not aware of function space variational inference, the paper is very hard to follow through. Perhaps sections 2.1 and 2.2 can be trimmed to accommodate some background on function-space VI.
	2.	Furthermore, related to the above point, though I understand why the term “function-space” might be appropriate, it gives an impression that authors are referring to “space of functions”. Then, it is mentioned that “the function space is typical of much lower dimension compared to the weight space”. However, the space of functions has infinite dimensions. Are the authors may be referring to the dimension of the range of functions? This is actually a bit confusing. If this is not the case, then the motivation of function space VI is not strong enough.
	3.	In Eq. 16, the authors note that while calculating $\hat{\mu_u}$ there is a stop-gradient operator in action. The same is not said for the step calculating $\hat{\Phi_u}$. I assume this means that the gradients are back-propagated through the SGD steps to update the pseudo-corsets. If yes, then I would like to see the effect of removing the stop-gradient operator while calculating $\hat{\mu_u}$. Further,  doesn’t this make the proposed method computationally expensive too? As is the case with BPC? However, if this is not the case, then this should be highlighted more explicitly.
	4.	The authors have shown the architecture generalization by changing the type of normalization layer for a single architecture. However, in my opinion, this is not good enough. In dataset summarisation literature, architecture generalization is often shown by changing the architecture itself. Hence, I would suggest bringing Tables 5 & 6 from the appendix to the main manuscript.
	5.	Do the authors make use of Differentiable Siamese Augmentation (DSA) [2]? This is not mentioned in the main text and can affect the result by large margins. If not, then I suggest the authors try it as this can improve the results as well.
	6.	I think the authors can also include a few basic Dataset Condensation methods in comparison for eg. GM [3], DM [4], DSA [2].
	7.	In Table 6 of the appendix, the authors note that there is performance degradation while using big architectures like ResNet. However, doesn’t this in some sense defy the motivation noted in the proposed work? Since, the proposed method operated in function space rather than the parameter space which (as the authors say) is lower dimensional, shouldn’t the performance be agnostic to architecture choice?
	8.	Few recent works might be relevant to mention in related works [5, 6, 7]


[1] T. G. Rudner, Z. Chen, Y. W. Teh, and Y. Gal. Tractable function-space variational inference in bayesian neural networks, 2022.
[2] B. Zhao and H. Bilen. Dataset condensation with differentiable siamese augmentation. In Proceedings of The 38th International Conference on Machine Learning (ICML 2021), 2021.
[3] B. Zhao, K. R. Mopuri, and H. Bilen. Dataset condensation with gradient matching. In International Conference on Learning Representations, 2021.
[4] Bo Zhao and Hakan Bilen. Dataset condensation with distribution matching. In Proceedings of the IEEE/CVF Winter Conference on Applications of Computer Vision, pages 6514–6523, 2023
[5] Xu, Yue, et al. "Distill Gold from Massive Ores: Efficient Dataset Distillation via Critical Samples Selection." arXiv preprint arXiv:2305.18381 (2023).
[6] Tiwary, Piyush, Kumar Shubham, and Vivek Kashyap. "Constructing Bayesian Pseudo-Coresets using Contrastive Divergence." arXiv preprint arXiv:2303.11278 (2023).
[7] Yin, Zeyuan, Eric Xing, and Zhiqiang Shen. "Squeeze, Recover and Relabel: Dataset Condensation at ImageNet Scale From A New Perspective." arXiv preprint arXiv:2306.13092 (2023).

Overall, the paper is definitely intriguing and excites the reader. However, the lack of background in the paper confuses the reader. Further, some points need to be clarified (see above).

---

> ### Author Rebuttal · Authors · 2023-08-10
>
> We sincerely appreciate your constructive comments. We respond to the individual comments below:
>
> **[W1, Q1]** Thank you for the valuable suggestion. We will consider reorganizing the paragraphs during the paper revision.
>
> **[Q2]** I agree with your point that the statement 'function space itself has a lower dimension compared to weight space' can be confusing. It seems better to rephrase it as 'functions have a lower dimension than weights,' as you suggested.
>
> **[Q3]** In fact, in the process of calculating $\hat\Psi$ in Equation 16, we have indicated the stop-gradient operator as **'sg.'** This implies that back-propagation is no longer applied to the parameters computed during SGD steps in our approach. The gradient of the pseudocoreset flows through the input $u$ in the calculation of $\hat\mu$. Moreover, it requires several tens of SGD steps to reach the MAP solution of the pseudocoreset, and an additional 30 SGD steps are used to obtain empirical variance. Storing gradients throughout all these processes would be computationally very expensive. This does not align with our intention to create a scalable BPC algorithm.
>
> **[Q4]** We will make sure to do that during the revision.
>
> **[Q5]** We agree with the statement that DSA [1] impacts the performance. Following the convention in the existing literature, we employed DSA in our approach and will make sure to mention it in the paper.
>
> **[Q6]** To the best of our knowledge, dataset distillation primarily focuses on achieving test performance matching, which can be seen as making the **point estimation** performance of parameters align as an objective. Accordingly, the evaluation of the distilled dataset is compared using SGD performance. On the other hand, methods like ours BPC aim to address how well **posterior distributions align.** Consequently, in our paper, we evaluate using the SGHMC sampling technique. Nevertheless, for the purpose of comparing with the relevant field, a comparison against the SGD performance of a few Dataset condensation methods yields the following [1,2,3].
>
> **< Test accuracies for CIFAR10 dataset >**
> | ipc | GM (SGD)       | DSA (SGD)      | DM (SGD)       | Ours (SGHMC)   |
> |-----|----------------|----------------|----------------|----------------|
> | 1   | 28.3 $\pm$ 0.5 | 28.8 $\pm$ 0.7 | 26.0 $\pm$ 0.8 | **35.5 $\pm$ 0.3** |
> | 10  | 44.9 $\pm$ 0.5 | 52.1 $\pm$ 0.5 | 48.9 $\pm$ 0.6 | **62.3 $\pm$ 0.3** |
> | 50  | 53.9 $\pm$ 0.5 | 60.6 $\pm$ 0.5 | 63.0 $\pm$ 0.4 | **71.2 $\pm$ 0.2** |
>
> **[Q7]** In the appendix, we have indicated that there is typically a performance degradation of the pseudocoreset in larger architectures such as ResNet. We believe this is primarily due to the limited dataset size used to evaluate the trained pseudocoreset, leading to increased overfitting in these larger architectures. This phenomenon is also commonly observed in existing dataset distillation methods. Our contribution, utilizing the function space, enables more scalable use of larger architectures like ResNet for the pseudocoreset training, and our method's performance on ResNet is superior to that of existing literature [4]. This reaffirms the validity of our approach.
>
> **[Q8]** We appreciate the valuable recommendations, and we will include the mentioned papers in the related work section during the revision.
>
> **[W2]** Please refer to our general responses [G2, G3].
>
> ---
> **References**
>
> [1] B. Zhao and H. Bilen. “Dataset condensation with differentiable siamese augmentation.” In Proceedings of The 38th International Conference on Machine Learning (ICML 2021), 2021.
>
> [2] G. Cazenavette, T. Wang, A. Torralba, A.A. Efros, J.Y. Zhu. “Dataset Distillation by Matching Training Trajectories.” CVPR, 2022.
>
> [3] Bo Zhao, Hakan Bilen. “Dataset Condensation with Distribution Matching.” WACV 2023, 2023.
>
> [4] R. Yu, S. Liu, and X. Wang. Dataset distillation: A comprehensive review, 2023.

---

> > ### Comment · Reviewer_d2wU · 2023-08-15
> > **Response on the rebuttal.**
> >
> > Thanks, Authors for clarification. Some of my questions are answered, but not all.
> >
> > (i) The computational complexity has to be quantified.
> >
> > (ii) Distillation methods that are compared are rather old (There are recent distillation methods such as MTT).
> >
> > (iii) I am still not convinced about the scalability of the approach with larger architectures.
> >
> > (iv) Performance of the method without the use of DSA needs to be measured and reported.

---

> > > ### Author Response · Authors · 2023-08-16
> > > **Response to the response**
> > >
> > > We appreciate your effort to review our paper and responses.
> > >
> > > **(i)** Our algorithm's iteration consists of training BPC to find the MAP solution, slightly advancing SGD steps to obtain the empirical covariance for eq.17, and calculating the final loss to update the pseudocoreset.
> > > In terms of memory considerations, as previously mentioned in Section 3.5 and general response [G1], FBPC excels. However, in terms of time, our method requires slightly more time because more SGD steps are needed to acquire the empirical covariance. However, when using FBPC-random, these steps can be reduced, trading off a slight decrease in performance for time savings.
> > > Mainly the time complexity is typically dominated by the SGD steps required to find the MAP solution. Finally, representing the time taken for one pseudocoreset update in wall clock time is as follows. We will include this discussion in the future revisions of the paper.
> > >
> > > **< wall-clock time (sec) for 1 step update (CIFAR10) >**
> > > | ipc               | 1               | 10              | 50               |
> > > |:-----------------:|:---------------:|:---------------:|:----------------:|
> > > | BPC-fKL | 1.04 $\pm$ 0.10 | 1.37 $\pm$ 0.13 | 2.59 $\pm$ 0.86  |
> > > | FBPC    | 1.5 $\pm$ 0.15  | 3.29 $\pm$ 0.51 | 8.38 $\pm$ 0.48  |
> > >
> > >
> > >
> > > **(ii)** We have conducted a performance comparison between our method, MTT [1], and FrePo [2] in the field of dataset distillation. MTT and FrePo present SGD performance from their respective papers, while our FBPC demonstrates SGHMC performance as outlined in our paper. When compared with other dataset distillation methods, we believe that our approach achieves performance comparable to the state-of-the-art DD method. Additionally, it is noteworthy that our method consumes significantly less memory than MTT, and it can also handle cases like Tiny-ImageNet ipc 50, which were not achievable using the FrePo approach.
> > > |               | ipc | MTT (SGD)  | FrePo (SGD) | FBPC (SGHMC) |
> > > |:-------------:|:---:|:----:|:-----:|:----:|
> > > | CIFAR10       | 1   | 46.3 | 46.8  | 35.4 |
> > > |               | 10  | 65.3 | 65.5  | 62.3 |
> > > |               | 50  | 71.6 | 71.7  | 71.2 |
> > > | CIFAR100      | 1   | 24.3 | 28.7  | 21.0 |
> > > |               | 10  | 40.1 | 42.5  | 39.7 |
> > > |               | 50  | 47.7 | 44.3  | 44.4 |
> > > | Tiny-ImageNet | 1   | 8.8  | 15.4  | 10.1 |
> > > |               | 10  | 23.2 | 25.4  | 19.4 |
> > > |               | 50  | 28.0 | -     | 26.4 |
> > >
> > >
> > > **(iii)** As indicated in the general response [G1], as the network size increases, the memory requirements for weight space BPC can become prohibitively large, rendering training infeasible. In contrast, our FBPC offers a distinct advantage, as it does not impose significant memory burdens even for larger architectures. Therefore, conducting experiments with FBPC is not hindered by memory constraints, allowing us to explore results on larger architectures.
> > >
> > > Indeed, we plan to conduct experiments on even larger architectures to provide a comprehensive understanding of our method's scalability. However, for new architectures, the creation of expert trajectories is a prerequisite. Given the potential time-consuming nature of this process, particularly for substantial architectures, we aim to share the results as soon as experimentation is completed.
> > >
> > > If there are any other concerns, please feel free to inform us. We appreciate your input and are committed to addressing any additional questions or considerations.
> > >
> > >
> > >
> > > **(iv)** We present results for BPC-fKL and FBPC without using DSA [3] and without any augmentation during training, and we will include these details in the paper. Interestingly, for the ipc 1 case in BPC-fKL, performance improved when DSA was not applied. However, in all other cases, it is evident that not using DSA leads to an average performance drop of approximately 4.7%. Moreover, even when training BPC without augmentation, we observe that function space BPC outperforms weight space BPC.
> > >
> > > **<BPC Performance with and without DSA>**
> > > | ipc              | 1                    | 10                   | 50                   |
> > > |------------------|----------------------|----------------------|----------------------|
> > > | BPC-fKL (no DSA) | **37.26 $\pm$ 1.65** | 50.48 $\pm$ 1.39     | 60.75 $\pm$ 0.26     |
> > > | FBPC (no DSA)    | 33.69 $\pm$ 2.73     | 55.07 $\pm$ 1.30     | 66.03 $\pm$ 0.21     |
> > > | FBPC (DSA)       | 35.45 $\pm$ 0.31     | **62.33 $\pm$ 0.34** | **71.23 $\pm$ 0.17** |
> > >
> > > ---
> > > **References**
> > >
> > > [1] G. Cazenavette, T. Wang, A. Torralba, A.A. Efros, J.Y. Zhu. “Dataset Distillation by Matching Training Trajectories.” CVPR, 2022.
> > >
> > > [2] Yongchao Zhou, Ehsan Nezhadarya, Jimmy Ba. "Dataset Distillation using Neural Feature Regression." NeurIPS, 2022.
> > >
> > > [3] Zhao et al. “Dataset condensation with differentiable siamese augmentation.” ICML, 2021.

---

> > > > ### Comment · Reviewer_d2wU · 2023-08-16
> > > > **Follow up**
> > > >
> > > > Thanks for the additional experiments. I'll increase my score.
> > > >
> > > > However, please improve the writing to reduce ambiguity.

---

### Official Review · Reviewer_9hB5 · 2023-07-06

**Soundness:** 3 good
**Presentation:** 2 fair
**Contribution:** 3 good
**Rating:** 7
**Confidence:** 4

**Summary:**

This paper introduces a novel approach called Function Space Bayesian Pseudocoreset (FBPC) for constructing Bayesian pseudocoresets for Bayesian Neural Networks. A Bayesian pseudocoreset is a compact synthetic dataset that summarizes essential information from a large-scale dataset and can be used as a proxy dataset for scalable Bayesian inference. Typically, the construction of a Bayesian pseudocoreset involves minimizing the divergence between the posterior conditioning on the pseudocoreset and the posterior conditioning on the full dataset. However, evaluating this divergence measure can be challenging, especially for models like deep neural networks with high-dimensional parameters.

In contrast to previous methods that construct and match coreset and full data posteriors in weight space (model parameter space), this proposed method operates directly in function space. By working directly in function space instead of weight space, several challenges such as limited scalability and multi-modality issues can be bypassed.

The method constructs variational approximations to the coreset posterior on function space by linearization and variational approximation to true posterior distributions. It then matches these approximations with full data posteriors also defined in function spaces.

Working directly in function spaces allows this approach to scale well even for large models where traditional weight-space approaches struggle computationally. Additionally, it does not constrain matching only specific architectures of neural networks but rather allows training with the equivalent of multiple architectures simultaneously while still achieving similar results.

Furthermore, using function-space matching improves out-of-distribution robustness compared to previous approaches based on weight spaces.Overall, experiments conducted demonstrate that using Function Space Bayesian Pseudocoresets leads to enhanced uncertainty quantification abilities compared to traditional methods based solely on weights or model parameters.

**Strengths:**

* The authors tackle a very interesting problem, extracting reliable posterior distributions out of BNNs to perform more precise inference. The Bayesian pseudocoreset approach allows for a simple but practical solution to this task.

* The paper is easy to follow and is very well written. The authors provide much or the context needed to understand the submission and the previous state-of-the-art.

* The function space approach is a very promising framework to conduct inference in BNNs and other models, and advances such as this one would help spreading its usage.

**Weaknesses:**

* While reading the paper, I could not help but miss some a comparison with the concept of inducing points, so common in sparse variational approximations for Gaussian Processes. From my point of view this is missing and would be interesting, since the proposed approach seems to be conceptually very strongly related to SVGPs, so somewhat including a comparison would be helpful.

* To avoid scalability issues regarding the Jacobian defined in the linearized approximation, the authors then approximate the posterior in function space using variational Gaussian distributions that then myst be fit. This can be scalable, although I fear it may induce strong bias to the final posterior estimate, since there is no guarantee that the posterior distribution would behave in this manner.

* I think that some contributions relevant to the topic at hand are left out from the related work section, specially in the function space variational inference paragraph in section 4. I suggest including [1,5] as context, maybe [6] as well for context. Moreover, in terms of approximating the posterior distribution from BNNs, all a-posteriori Laplace approximation-based approaches are missing, such as [4] and [7]. These last two are not crucial for the topic at hand, although could serve to complete further the discussion.

* I would expect a more detailed experimental phase, characterizing further the properties of the model. I would suggest complementing the results with some regression results. Moreover, it would be interesting to compare the resulting predictive distribution in each case with other models such as HMC, MFVI or other function space methods s.a. [1,2,3,5]. Moreover, it would be interesting to know how the distributions obtained compare against a-posteriori methods s.a. [4].

* I would include some scalability study for the method, specially in terms of the size and dimensionality of the dataset.

_Note:_ I condition my final evaluation on the submission to the fact that the authors address these issues. Were this not the case, I may consider revising the score.

---

**References**:

[1] Rodrı́guez-Santana, S., Zaldivar, B., & Hernandez-Lobato, D. (2022, June). Function-space Inference with Sparse Implicit Processes. In International Conference on Machine Learning (pp. 18723-18740). PMLR.

[2] Rudner, T. G., Chen, Z., Teh, Y. W., & Gal, Y. (2022). Tractable function-space variational inference in Bayesian neural networks. Advances in Neural Information Processing Systems, 35, 22686-22698.

[3] Ma, C., & Hernández-Lobato, J. M. (2021). Functional variational inference based on stochastic process generators. Advances in Neural Information Processing Systems, 34, 21795-21807.

[4] Deng, Z., Zhou, F., & Zhu, J. (2022). Accelerated Linearized Laplace Approximation for Bayesian Deep Learning. Advances in Neural Information Processing Systems, 35, 2695-2708.

[5] Ma, C., Li, Y., and Hernández-Lobato, J. M. (2019). “Variational implicit processes”. In: International Conference on Machine Learning, pp. 4222–4233.

[6] Fortuin, V. (2022). Priors in bayesian deep learning: A review. International Statistical Review, 90(3), 563-591.

[7] Antorán, J., Janz, D., Allingham, J. U., Daxberger, E., Barbano, R. R., Nalisnick, E., & Hernández-Lobato, J. M. (2022, June). Adapting the linearised laplace model evidence for modern deep learning. In International Conference on Machine Learning (pp. 796-821). PMLR.


**Questions:**

* What role does the forward KL divergence play in constructing the Bayesian pseudocoresets, and why would you prefer this to well-defined distance measurement or any other divergence measure? Does the assymetry of the KL divergence bias the results in any significative manner that one should be aware of when using the method?

* Have you performed any checks comparing the usage of the full Jacobian [lines 151 and 152] with the case where you induce the variational approximateion to the finite-dimensional distributions of the posteriors in function space? I agree that this cannot be done in general, specially for big BNNs where the computational cost of computing the Jacobian is high. However, tests could be performed in small BNNs to check whether this assumption holds in those simpler instances. If you consider this to not be necessary, please argue why.

* Robustness to the selection of BNN architecture is an important point in the submission. While I agree that the results obtained in this regard seem to point in that direction, I would also argue that differences may be observed if it were not for the approximations regarding the linearization of the system and the imposed Gaussian variational form. Is there any reason to think otherwise?

* Why would this approach be preferred over those that obtain predictive distributions with a-posteriori approximations s.a. [4] and [7]?

**Limitations:**


* The Bayesian pseudocoreset is only constructed comparing Gaussian distributions, which may be insufficient to express different posteriors induced by the data.

* The experimental part of the submission could be improved.

* There are no studies on the scalability of the method.

---

> ### Author Rebuttal · Authors · 2023-08-10
>
> We sincerely appreciate your constructive comments. We respond to the individual comments below:
>
> **[W1]** Thanks for your insightful comment. We will include the relevant discussion in the paper. As you pointed out, inducing points in Stochastic Variational Gaussian Processes and the functional Bayesian pseudo coresets in BNN are similar in their purpose.
> * SVGP : In the context of Stochastic Variational Gaussian Processes, the process of optimizing the inducing points denoted as $Z=${$z_1,...,z_n$} involves maximizing the ELBO in order to make a variational Gaussian distribution $q(f_{tr},f_z)$ well approximate the posterior distribution $p(f_{tr},f_z|y_{tr})$. This variational distribution is composed of two parts: $p(f_{tr}|f_z)$, which represents the Gaussian posterior distribution, and $q(f_z)$, which is the Gaussian variational distribution. During this optimization, we focus on training the inducing points $Z$ as well as the mean and variance of the variational distribution $q(f_z)$. The goal of this optimization is to create a good approximation of the posterior distribution $p(y_{te}|x_{te}, D_{tr})$ during the inference process, all while keeping the computational cost low.
> * FBPC : As outlined in section 3.2, the formulation of FBPC involves directly minimizing the divergence between function space posteriors, specifically $D(\nu_x, \nu_u)$.
>
> To sum up, while they do share some similarity in that they introduce a set of learnable pseudo data points, they are fundamentally different in their learning objectives. SVGP is interested in approximating the full data posterior through the inducing points, while ours aims to make the pseudocoreset posterior as close as possible to the full data posterior.
>
> **[W2]** Please refer to our general response [G5].
>
> **[W3]** We appreciate the valuable recommendations, and we will include the mentioned papers in the related work section during the revision.
>
> **[W4]** Please refer to our general response [G2] for the regression experiment. As you mentioned, a comparison with other methods like HMC would indeed be interesting. Analyzing the differences between the predictive distributions obtained from a-posteriori methods and those obtained from BPC could be an engaging avenue for future work.
>
> **[W5]** Please refer to our general response [G1].
>
> ---
> **[Q1]** There are two major advantages in using the forward KL divergence.
> * Firstly, when compared to the reverse KL divergence, which has a mode capturing property, forward KL minimization favors solutions covering the entire target distribution. This characteristic makes it more suitable for Bayesian pseudocoreset construction, as evaluating model performance through Bayesian model averaging (BMA) allows better consideration of diversity over the entire distribution rather than focusing on individual modes. This advantage is also acknowledged in [1], where they compared three divergence measures—forward KL, reverse KL, and Wasserstein distance—and demonstrated the superiority of forward KL.
> * Secondly, another advantage of using forward KL is the ease of computing the objective. While there are many other well-defined metrics available, for BPC optimization, it is crucial to have the gradient of the divergence between two posteriors with respect to BPC. We were able to derive the derivative for forward KL through proposition 3.1, which enabled us to compute FBPC.
>
> Exploring the possibility of similar derivations for other divergences could be a promising topic for future work.
>
> **[Q2]** Please refer to our general response [G4].
>
> **[Q3]** Thank you for the insightful comment. While it may raise similar concerns as you mentioned, Table 6 of the Appendix demonstrates the simultaneous learning of pseudocoresets for two entirely different architectures: CNN and ResNet. This result illustrates that our method is capable of accommodating various architectures, and it is not solely reliant on the linearization and Gaussian variational form. Instead, we attribute this capability to the effect of similar function posteriors mentioned in Section 3.4.
>
> In addition, we explored using a diagonal covariance to achieve a more refined approximation of the posterior in weight space BPC. However, as shown below, this approach still did not yield significant benefits in terms of architecture generalization. We believe that matching the function posterior distribution is more helpful for enhancing architecture robustness, rather than solely focusing on how well we approximate the posterior in weight space.
>
> | BPC-fKL  | LN | BN | GN | IN |
> |----------|-----------|-----------|-----------|-----------|
> | Accuracy | 40.06     | 46.74     | 40.96     | 53.07     |
>
> **[Q4]** As you mentioned, there are indeed various methods to obtain predictive distributions, including a-posteriori approximations [2,3]. It is worth noting that our proposed method, utilizing the pseudocoreset, can also be applied in conjunction with a-posteriori methods to further reduce computational burden while obtaining predictive distributions.
>
> Additionally, our approach offers the advantage of obtaining a **small condensed dataset** which can have versatile applications. This condensed dataset can be leveraged as memory data in transfer or continual learning settings, and it may hold value as a learnable prior in Bayesian settings. Utilizing a pseudocoreset as a learnable prior, rather than storing weights directly, opens up interesting possibilities for future work. Investigating the feasibility of using small datasets as a learnable prior represents a promising avenue for further research.
>
> ---
> **References**
>
> [1] Kim et al. “On Divergence Measures for Bayesian Pseudocoresets.” NeurIPS 2022.
>
> [2] Daxberger et al. “Laplace Redux -- Effortless Bayesian Deep Learning.”  NeurIPS 2021.
>
> [3] Deng, Z., Zhou, F., & Zhu, J. “Accelerated Linearized Laplace Approximation for Bayesian Deep Learning.” NeurIPS 2021.

---

> > ### Comment · Reviewer_9hB5 · 2023-08-20
> > **Brief response to the rebuttal**
> >
> > I would like to thank the authors for the extensive work put forth in the rebuttal of all the reviews. I consider there is a good amount of work behind the responses, and I appreciate it deeply.
> >
> > As far as I am concerned, I consider my questions and doubts mostly addressed. I would suggest the authors augmenting the original text with some of the rebuttals' contents in order to provide a more complete explanation: for example, the discussion around SVGPs, the choice of divergences or the usage of _a posteriori_ techniques may result enriching.
> >
> > I will maintain my score since I think it's suitable, but I am in fact even more positive about this submission than earlier. Thanks again!

---

### Official Review · Reviewer_sWjp · 2023-07-06

**Soundness:** 3 good
**Presentation:** 3 good
**Contribution:** 2 fair
**Rating:** 4
**Confidence:** 4

**Summary:**

This paper combines the works of Kim et al. (2022) and Rudner et al. (2022) to propose a Bayesian coreset learning method based on function space variational inference. The authors find that this leads to coresets with improved accuracy/NLL compared to (Kim et al., 2022) on CIFAR and Tiny-ImageNet benchmarks as well as improved robustness against data corruption on CIFAR10. Finally, the authors argue that their method allows for improved incorporation of different architectures in coreset learning based on an improved generalization across architectures with different types of normalization layers for CIFAR10.

Generally, I found the paper quite incremental, although I appreciate that clear credit is given to prior work. The use of terminology is also rather loose, not any ad-hoc construction of a distribution over the weights constitutes an approximation to a Bayesian posterior. Finally, while performance is improved over that of Kim et al. (2022), in absolute terms it is still really far off that of training on the full dataset. I would really want to see that the coreset performance converges to that of training on the full dataset, so at this point I would argue for rejection.

**Strengths:**

* The method is new and described clearly
* Clear credit is given to prior work that the method is based on
* Performance is better than that of the baseline
* The architecture generalization supports/motivates the use of function space inference

**Weaknesses:**

* The method as such seems really incremental, essentially it plugs the functional variational inference of Rudner et al. (2022) into the coreset learning and posterior “approximation” method of Kim et al. (2022)
* That being said, calling a Gaussian distribution constructed from an optimization trajectory by taking the mean and variance of the parameters and calling it a variational distribution seems like a bit of an abuse of terminology. Not every distribution approximates the posterior. Further, this approach seems mostly equivalent to SWAG (Maddox et al., 2019), so this should be cited (and any potential difference discussed if applicable).
* The absolute accuracies even for the largest coresets are still really far off what we would get when training on the full dataset. So the method seem quite far away from being useful to practitioners.


**Questions:**

* Does the gap with full training close when further increasing the coreset size?
* The experimental section states that BPC-fKL was not possible on Tiny-Imagenet with 50 coreset points: why is that exactly? It would be useful to have some more details on memory consumption/runtime and the corresponding scaling behavior with coreset size/dateset size/dimension/number of parameters for the different methods. Reporting. some empirical numbers from the experiments on VRAM use etc could be helpful.

**Typos/minor:**
* l65-66: remove “number of”?
* l106: “the means” -> “means”
* l238: “we” -> “We”
* l247: “refer the appendix” -> “refer to the appendix”
* eq (6): $\Sigma_u, \Sigma_x$ appear for the first time here, but are only defined after eq (12).
* I found the name for the FBPC-random baseline really confusing vs the random coreset points baseline, perhaps calling FBPC-iso or FBPC-isotropic would be more descriptive?
* The references could use another round of editing, please ensure that all citations have a venue (missing e.g. for 1, 2, 6, 15, 22, 27), and that title capitalization and venue names are consistent.
* The abstract is a bit verbose while providing fairly limited specifics. I would suggest cutting back the introductory first half — unfamiliar readers won’t learn much from this anyway and experts know all of this already.


**Limitations:**

I would have liked to see some discussion on the large gap to training on the full dataset.

---

> ### Author Rebuttal · Authors · 2023-08-10
>
> We sincerely appreciate your constructive comments. We respond to the individual comments below:
>
> **[W1]** We agree that our method extends the discussion in Kim et al. [1] to function space posterior. However, we want to emphasize additional significant contributions in our work.
> * Firstly, we have demonstrated that obtaining a lower-dimensional function space matching for Bayesian pseudocoreset is easier than complex weight posterior matching, which is a novel finding.
> * Secondly, we propose scalable methods (Section 3.2, 3.3) for matching function space posterior, enabling computations in high-dimensional BNNs. Our algorithm that constructs variational posteriors directly in function spaces via trajectory statistics is significantly different from the previous function-space VI methods [2, 3] where they construct variational posteriors as pushforwards of weight-space variational posteriors.
> * Furthermore, by leveraging the similarity of function space across different architectures, our method can create architecture-robust BPC.
>
> These unique contributions distinguish our paper from others in the field. We sincerely appreciate your consideration of these points.
>
> **[W2]** Thank you for the valuable feedback. We will carefully consider the replacement regarding the terminology 'variational distribution' in our paper during the revision period. Additionally, we have identified the similarities between our method for creating posterior distributions and the SWAG method [4]. We will add this discussion to the related work section of the paper. But again, please note that while SWAG considers collecting statistics on weight space trajectories, ours constructs statistics in **function spaces**, which makes ours more suitable and scalable for pseudocoreset construction.
>
> **[W3]** Thank you for the thoughtful comment. It is true that there remains a gap between the performance achievable by training large-size models on full data and the results obtained through our approach. We have also recognized and contemplated this issue, and that led us to propose the FBPC method, which addresses the scalability problem of Bayesian pseudocoresets and allows us to extend the models and datasets to sizes that were not easily achievable with existing Bayesian pseudocoreset methods.
> Through FBPC, we successfully scaled our approach to handle larger models and datasets, demonstrating its potential in bridging the performance gap further. We agree that future research should continue in this direction, and we believe that our method serves as a valuable starting point for such endeavors.
>
> ---
> **[Q1]** Yes, that's correct. When we increased the coreset size to 100, we obtained an accuracy of 73.45, and when increased to 200, the accuracy reached 74.80. It is evident that the performance increases as the size is expanded.
>
> **[Q2]** As the coreset size, ipc, and network size increase, memory usage also increases, and the memory burden may make conducting experiments in our current environment impractical. Specifically, BPC-fKL requires Monte Carlo samples corresponding to the parameter dimension size, whereas FBPC only needs samples from the function space dimension, resulting in a significant difference in memory usage.
>
> However, during the rebuttal period, we discovered that by reducing the pseudocoreset batch size during training, it becomes feasible to run experiments on a 3090 GPU. With a batch size of 100 during training, we achieved performances of accuracy 22.18 $\pm$ 0.32 and NLL 4.65 $\pm$ 0.02 where our FBPC achieved performances of accuracy 26.43 $\pm$ 0.31 and NLL 4.30 $\pm$ 0.05.
>
> For scalability study of our method, please refer to our general response [G1].
>
> **Typos/minor** Thank you for finding them and we will correct them.
>
> ---
>
> **References**
>
> [1] Balhae Kim, Jungwon Choi, Seanie Lee, Yoonho Lee, Jung-Woo Ha and Juho Lee. “On Divergence Measures for Bayesian Pseudocoresets.”  In Advances in Neural Information Processing Systems 36 (NeurIPS 2022), 2022.
>
> [2] T. G. J. Rudner, G. Chen, and Y. Gal. “Rethinking function space variational inference in
> Bayesian neural networks.” 3rd Symposium on Advances in Approximate Bayesian Inference, 2020.
>
> [3] Tim G. J. Rudner, Zonghao Chen, Yee Whye Teh and Yarin Gal. “Tractable Function-Space Variational Inference in Bayesian Neural Networks.”  In Advances in Neural Information Processing Systems 36 (NeurIPS 2022), 2022.
>
> [4] Wesley J. Maddox, Timur Garipov, Pavel Izmailov, Dmitry Vetrov and Andrew Gordon Wilson. “A Simple Baseline for Bayesian Uncertainty in Deep Learning.”  In Advances in Neural Information Processing Systems 33 (NeurIPS 2019), 2019.

---

> > ### Comment · Reviewer_sWjp · 2023-08-15
> >
> > Thank you for your detailed and constructive response. I had a closer look through some of the related works provided by Rd2wU and the results on cifar10 at least seem to be in line with the literature (it might be worth citing a couple of these numbers), so I would say that performance is not strictly a barrier to acceptance and I will increase my score. However, I do still find the work for the most part very incremental, so I will only go up by 1 point.

---

### Author Rebuttal · Authors · 2023-08-10

We express our gratitude to all the reviewers for their valuable and insightful feedback. They have acknowledged the clarity in our method description (R-sWjp, R-9hB5) as well as our coverage of relevant prior works (R-9hB5). The reviewers have also recognized the paper's strong motivation and the significance of addressing an engaging problem (R-sWjp, R-9hB5, R-d2wU, R-jYGC). Moreover, they have emphasized that our proposed FBPC method demonstrates superior performance compared to other BPC approaches (R-sWjp, R-d2wU, R-jYGC), highlighting its practicality in Bayesian neural network inference (R-9hB5, R-jYGC). The reviewers have also highlighted our contributions, specifically our achievements in a multi-architecture setting (R-d2wU) and the empirical estimation of function space posterior (R-jYGC).

We sincerely appreciate all the constructive comments and we provide general responses addressing questions that were commonly raised by the reviewers below:

**[G1] (Scalability study)** In our paper, we proposed a more scalable BPC construction algorithm by focusing on distribution matching in the function space rather than the weight space. Therefore, we can effectively and efficiently create BPC even as the number of parameters increases. To compare how scalable our approach is compared to posterior matching in the weight space, we measured GPU memory usage corresponding to the number of parameters, and the results are as follows. As shown in the table, the memory usage for weight space BPC significantly increases as the number of parameters grows, while FBPC operates very efficiently.
Additionally, the coreset ipc increases memory usage proportionally to its size. We believe the same principle applies to dataset size as well.

**<GPU memory usage (GB) (CIFAR10)>**
|   ipc10           | LeNet           | ConvNet         | ResNet18        |
|--------------|-----------------|-----------------|-----------------|
| # Parameters | $6.2 \times 10^4$ | $3.2 \times 10^5$ | $1.1 \times 10^7$ |
| FBPC         | 0.02            | 0.32            | 2.56            |
| BPC-FKL      | 0.11            | 3.17            | 12.18           |

|  ipc       | 1    | 10   | 50    |
|---------|------|------|-------|
| FBPC    | 0.04 | 0.32 | 1.59  |
| BPC-fKL | 0.41 | 3.17 | 15.59 |

**[G2] (More experiments)** Our paper introduces a scalable BPC construction method for BNN classification. We extended our exploration to regression, using the yacht dataset with (308, 6) dimensions and an MLP with one hidden layer. The test mean squared error (MSE) for different coreset sizes is as follows. This confirms our method's effectiveness in regression.

The 'Random' method selected samples randomly from real datasets, potentially including outliers. As coreset size increases, the trained FBPC's test MSE approaches 0.0647, matching the test MSE from training on the entire dataset.

|        | 10     | 20     | 30     | 50     | 100    |
|--------|--------|--------|--------|--------|--------|
| Random | 0.3424 | 0.5335 | 0.2757 | 0.1230 | 0.0931 |
| FBPC   | 0.3148 | 0.2541 | 0.1434 | 0.0854 | 0.0829 |
| FBPC (Jacobian) | 0.3156 | 0.2460 | 0.1413 | 0.0872 | 0.0825 |

**[G3] (A lot of approximations)** To understand the impact of these approximations, we conducted regression task experiments on small BNNs, where it was possible to obtain the full Jacobian." As seen in the table from [G2], the results were not significantly different from ours. Analyzing the effects of each approximation in larger models and researching more refined and scalable approximation methods would be a valuable direction for future research.


**[G4] (Possibility of the usage of the full Jacobian)**
Incorporating the full Jacobian in computations is challenging, especially for our pseudocoreset updates that involve frequent calculations of equations 12 and 13. Computing $\Sigma_x$ and $\Sigma_u$ requires inverting the Hessian matrix, often approximated using the Fisher information matrix. For the full dataset, this becomes computationally demanding, and for the coreset, small size leads to high variance with the standard Fisher approximation. These challenges persist even for small BNNs.

Initially, we attempted several approximations for FBPC, especially for diagonal approximations of $\Sigma_x$ and $\Sigma_u$, which resulted in instability and hindered convergence. Employing empirical variance proved more stable for FBPC computation. For instance, using Jacobian and Fisher Information Matrix for a diagonal covariance in the CIFAR-10 ipc10 setting resulted in 56.97% accuracy for FBPC, which was less favorable than our method. This approach introduces additional hyperparameters, complicating optimal settings identification.

**[G5] (Concerns of Gaussian approximation)**
As highlighted by Reviewer 9hB5, there's no assurance that the actual posterior distribution would perfectly align with the variational Gaussian distribution used in function space approximation. Nonetheless, it's crucial to recognize that approximations are required for effective objective computation.

Employing linearization and variational Gaussian distributions in function space is a practical strategy to manage scalability and enable feasible computations [1]. Although this approximation introduces some bias, it's a necessary compromise to address the challenges of high-dimensional models.

In the context of our work, exploring more sophisticated techniques, such as normalizing flows [2,3], to design more complex posterior distributions is indeed a promising direction for future research.

---
**References**

[1] Tim G. J. Rudner, Zonghao Chen, Yee Whye Teh and Yarin Gal. “Tractable Function-Space Variational Inference in Bayesian Neural Networks.”  NeurIPS 2022.

[2] Kristiadi et al. “Posterior Refinement Improves Sample Efficiency in Bayesian Neural Networks.” NeurIPS 2022.

[3] Chen et al. “Bayesian inference via sparse Hamiltonian flows.”  NeurIPS 2022.

---

### Decision · Program_Chairs · 2023-09-21

**Decision:**

Accept (poster)

**Comment:**

This paper presents a novel pseudocoreset construction method for a relevant pushforward of the posterior in certain models, instead of the raw parameter posterior as usual. The reviewers found the writing generally clear, the method interesting, and the experiments compelling. The paper is slightly incremental given Rudner et al, and involves quite a few approximations. Note that there are some minor technical issues in a few spots (e.g. lines 126/127, where densities on function spaces are used but not well-defined; 172/173 unclear claim about the dimension of function spaces, etc). Please address these points as well as the points raised during the discussion for the camera ready.